# Learning to Factorize
# Spatio-Temporal Foundation Models

**Siru Zhong**[1], **Junjie Qiu**[1], **Yangyu Wu**[1], **Xingchen Zou**[1],
**Zhongwen Rao**[2], **Bin Yang**[3], **Chenjuan Guo**[3], **Hao Xu**[2,*], **Yuxuan Liang**[1,*]
[1]The Hong Kong University of Science and Technology (Guangzhou),
[2]Huawei 2012 Laboratories, [3]East China Normal University
{szhong691,ywu494,xzou428}@connect.hkust-gz.edu.cn,
{jayjunjieqiu,yuxuanliang}@hkust-gz.edu.cn,
{raozhongwen,xuhao77}@huawei.com, {byang,cjguo}@dase.ecnu.edu.cn

## Abstract

Spatio-Temporal (ST) Foundation Models (STFMs) promise cross-dataset generalization, yet joint ST pretraining is computationally costly and struggles with domain-specific spatial correlations. To address this, we propose FactoST, a factorized STFM that decouples universal temporal pretraining from ST adaptation. The first stage trains a space-agnostic backbone via multi-task learning to capture multi-frequency, cross-domain temporal patterns at low cost. The second stage attaches an lightweight adapter that rapidly adapts the backbone to specific ST domains via metadata fusion, interaction pruning, domain alignment, and memory replay. Extensive forecasting experiments show that in few-shot settings, FactoST reduces MAE by up to 46.4% versus UniST, uses 46.2% fewer parameters, achieves 68% faster inference than OpenCity, and remains competitive with expert models. This factorized view offers a practical, scalable path toward truly universal STFMs.

## 1 Introduction

Spatio–temporal (ST) data capture how signals evolve over time across complex spatial structures—such as traffic speeds on road networks, air-pollution levels from citywide sensors, or electricity loads at substations. Modeling ST data is fundamental to forecasting and decision support across science, engineering, and society, enabling proactive anticipation and intervention [77, 13, 58, 73].

In deep learning practice, Spatio-Temporal Graph Neural Networks (STGNNs) are the de facto workhorse for modeling such data [58, 39, 47, 27, 28], as depicted in Figure 1(a). Given the intricate nature of jointly learning spatial and temporal dependencies, early attempts *decompose* the learning problem into two complementary components: (i) a recurrent [32, 6, 46], convolutional [68, 64, 30], or attention-based module [34, 72, 18] that extracts *temporal dependencies* from each location's history, and (ii) a Graph Neural Network (GNN) [29] that propagates information along edges to capture *spatial correlations* among locations [32, 68, 64, 57, 41]. This design yields strong inductive bias, parameter efficiency, and state-of-the-art performance on a wide range of benchmarks.

Inspired by the transformative impact of Foundation Models (FMs) in language [44] and vision [5], researchers have recently begun to explore **STFMs** [36, 17, 23, 14]. The core idea is simple – *Pretrain a single model on diverse ST corpora (e.g., climate, traffic, energy) and adapt it to unseen datasets in a zero-shot or few-shot fashion*, as shown in Figure 1(b). Such cross-domain pretraining equips STFMs with broad cross-dataset spatio-temporal knowledge and generalization beyond single-dataset scopes, often outperforming task-specific STGNNs when labeled data is scarce [40, 69, 33, 70].

---

*Corresponding author

39th Conference on Neural Information Processing Systems (NeurIPS 2025).

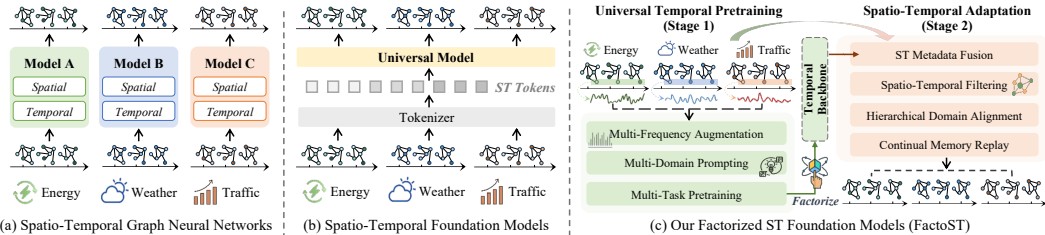

Figure 1: Evolution of ST modeling: (a) Traditional coupled STGNN design; (b) Joint ST pretraining in STFMs with tokens from different space and time; (c) FactoST's factorized paradigm.

Nevertheless, training an STFM at scale presents two pronounced obstacles. ***First***, spatial correlations differ dramatically across domains. For instance, the adjacency structure of a power grid differs greatly from urban road topology networks, making it difficult for a monolithic model to internalize all possible patterns; neighbouring air-quality stations in Beijing exhibit short-range diffusion dynamics, whereas tele-connection effects dominate climate indices across the Pacific Ocean. ***Second***, existing STFMs [69, 33] mostly rely on the paradigm of jointly learning spatial and temporal dependencies for hundreds, thousands, or even millions of locations, which is computationally expensive; memory and time footprints grow quadratically with sequence length or graph size in these architectures.

In this paper, we address these challenges by *factorizing STFM learning into two lightweight stages* and introduce **FactoST**, a new paradigm that decouples universal temporal learning from domain-specific ST adaptation. Generally, temporal patterns (such as seasonality, trends) share a common 1-D structure across domains, learnable once; spatial correlations hinge on domain-specific graphs with sizes and semantics, often requiring tailored reasoning. FactoST exploits this asymmetry: *it first distils the simpler temporal dynamics across domains, then attaches a compact adapter that injects the richer, domain-specific spatial knowledge*. Conceptually, FactoST can be seen as an "STGNN" in the era of FMs, reinstating spatio-temporal factorization at scale (see Figure 1(a)→(c)).

**Universal Temporal Pretraining (UTP):** The first stage aims to learn general temporal knowledge (e.g., periodicity) across diverse domains. To achieve this, we pretrain a purely temporal backbone on large-scale, cross-domain ST data, deliberately omitting any spatial modules. Multi-frequency augmentation is utilized to encourage the model to align multi-frequency information across scales, while domain-aware prompts guide it in encoding task-specific context without explicit spatial graphs. Overall, this stage is graph-agnostic, lightweight, and highly scalable with strong generalizability.

**Spatio-Temporal Adaptation (STA):** For a target dataset, the second stage freezes or fine-tunes the UTP backbone and attaches a compact adapter that injects spatial awareness and domain specificity in one streamlined pass. The adapter first enriches temporal features with learnable spatio-temporal identifiers, making each token location- and time-aware. It then adaptively modulates these features by computing three low-rank affinities—quantifying how strongly each token aligns with its static spatial embedding, temporal calendar embedding, and lagged historical context—and fusing them into dynamic per-token weights. Finally, hierarchical soft prompts align representations between the pretraining and target domains at both layer and token levels, while a small replay buffer periodically resurfaces earlier sequences to stabilize training and prevent catastrophic forgetting. By seamlessly weaving these components into a single stage, STA endows the universal temporal features learned in UTP with just enough ST reasoning to excel in new tasks, achieving this with minimal computation and memory overhead. We summarize our technical contributions as follows:

- We propose a two-stage factorized paradigm for STFMs that decouples Universal Temporal Pretraining (UTP) from Spatio-Temporal Adaptation (STA), enabling efficient learning and adaptation while preserving strong temporal capabilities without the need for costly joint ST pretraining.

- We introduce key innovations in both stages: (1) In UTP, we leverage *multi-frequency augmentation*, *multi-domain prompting*, and *multi-task pretraining* to learn universal temporal patterns; and (2) In STA, we introduce ST metadata fusion (STMF) for spatial-aware feature alignment; ST filtering (STF) for sparse interaction modeling; hierarchical domain alignment (HDA) to bridge domain gaps; and continual memory replay (CMR) to mitigate knowledge forgetting.

- Extensive experiments show that FactoST outperforms existing STFMs by up to 46.4% in MAE under few-shot settings, while reducing parameter count by 46.2% and inference latency by 68%. It remains competitive with domain-specific expert models, even without architecture customization.

## 2 Preliminary

### 2.1 Formulation

**Definition of Spatio-Temporal Data.** We define *Spatio-Temporal (ST) data* $\mathcal{D} = (\mathcal{X}, A, \mathbf{M})$ as a sequence of multivariate observations recorded regularly at a fixed set of spatial locations. Formally, let $V = \{v_1, \ldots, v_N\}$ denote $N$ nodes, e.g., traffic sensors, grid cells, and weather stations. Each node provides a $D$-dimensional feature vector at every time step over a horizon of length $L$, forming a tensor $\mathcal{X} \in \mathbb{R}^{N \times L \times D}$. Spatial interactions are represented by an (often sparse) *adjacency matrix* $\mathbf{A} \in \mathbb{R}^{N \times N}$, whose entries encode physical distance, functional similarity, or learned affinity. Many real-world datasets also carry *node metadata* $\mathbf{M} = \{m_i\}_{i=1}^{N}$, such as geo-coordinates or land-use types that supply auxiliary spatial context. This formulation also subsumes several commonly used representations, including multivariate time series [79, 80, 12, 56] and ST raster data [74, 69, 4, 19].

**Goal of Spatio-Temporal Foundation Models (STFM).** Most existing STFMs centre on ST forecasting, as accurate future prediction is the cornerstone for a majority of ST applications, from traffic management to weather early-warning systems [36, 69, 17]. STFMs therefore seeks to learn a representation function $\Phi(\cdot)$ that converts a large, cross-domain corpus of ST datasets into a general-purpose representation $\mathbf{H} = \Phi(\mathcal{D}_1, \ldots, \mathcal{D}_{n_d})$, where $n_d$ is the number of ST datasets. The key requirements for such a representation are: i) *Versatility*: It should support a broad spectrum of downstream forecasting tasks, including both short-term and long-term forecasting across diverse domains. ii) *Efficiency*: Adapting to a new task or domain must involve only a lightweight prediction head and minimal fine-tuning, while still matching or surpassing fully retrained, task-specific models.

### 2.2 Related Work

**Spatio-Temporal Graph Neural Networks (STGNNs).** STGNNs are the de-facto backbone for learning representations from complex ST data, powering tasks that range from ST forecasting and anomaly detection to classification and imputation [52, 27, 28, 15, 2]. Early STGNNs often factorize the learning problem into two complementary components: (i) temporal modules (e.g., RNNs [32, 24], TCNs [64, 65]) to extract sequential patterns at individual nodes, and (ii) spatial modules (e.g., GCNs [29, 78], GATs [72, 51]) to propagate information across graph edges. This factorized design, implemented either in stacked form [68, 76] or as tightly coupled pipelines [32], has proven effective across diverse domains such as traffic forecasting [6, 26], energy industry [1, 53] and environmental applications [35, 11]. Building on these foundations, recent work explores *self-supervised* objectives (e.g., contrastive or generative pretext tasks) to extract domain-agnostic ST features without dense labels [38, 25]. Transformer-style STGNNs further extend receptive fields with a self-attention mechanism while retaining the ST factorization [66, 37]. Despite these advances, most existing models are still trained from scratch for each dataset, which *limits their cross-domain reuse and falls short of "training once, adapt everywhere"*.

**Spatio-Temporal Foundation Models.** Recent efforts have explored STFMs that learn universal representations through cross-domain pretraining [36, 17]. Notable examples include UNIST [69] and OPENCITY [33], which apply transformer-based architectures to large-scale traffic data. As shown in Figure 1(b), UNIST tokenizes ST data into a sequence of ST tokens for Transformer-based learning and prompt-based adaptation. OPENCITY integrates Transformers with GNNs for flexible graph modeling, yet its tightly coupled architecture demands domain-specific pre-processing (e.g., road networks) and is prone to overfitting to particular spatial dependencies. Both models rely on expensive joint ST pretraining, leading to suboptimal performance and high computational cost. In contrast, time series foundation models like TIMESFM [10] and CHRONOS [3] achieve strong cross-domain generalization through purely temporal pretraining, but they lack any spatial awareness.

Our factorized framework bridges this gap: it first learns universal temporal patterns in a scalable manner, then injects lightweight spatial adapters for rapid ST adaptation, achieving both versatility and efficiency without the heavy cost of joint pertaining on both spatial and temporal dimensions.

Table 1: Qualitative comparison of STGNNs, STFMs, and our factorized STFMs (FactoST).

| Aspect | STGNNs | Existing STFMs | FactoST (ours) |
|---|---|---|---|
| **Training strategy** | Train from scratch on each dataset | Joint ST pretraining on massive corpora | **Factorized**: UTP + STA |
| **Temporal modeling** | RNN / TCN / self-attention | Transformer on $(N \times T)$ tokens | Freq. aug. + domain prompts + Transformer on $T$ tokens |
| **Spatial modeling** | GCN / GAT tied to one graph | Transformer + GNN or grid tokenization | Pluggable lightweight adapters, dynamic edge pruning |
| **Computational cost** | Moderate (per-domain repeat) | Very high (joint ST pretraining) | Low-to-moderate (no spatial cost in UTP, small adapters in STA) |
| **Cross-domain reuse** | Minimal (bound to one graph) | Partial (tied to graph modeling) | High (graph-agnostic backbone) |

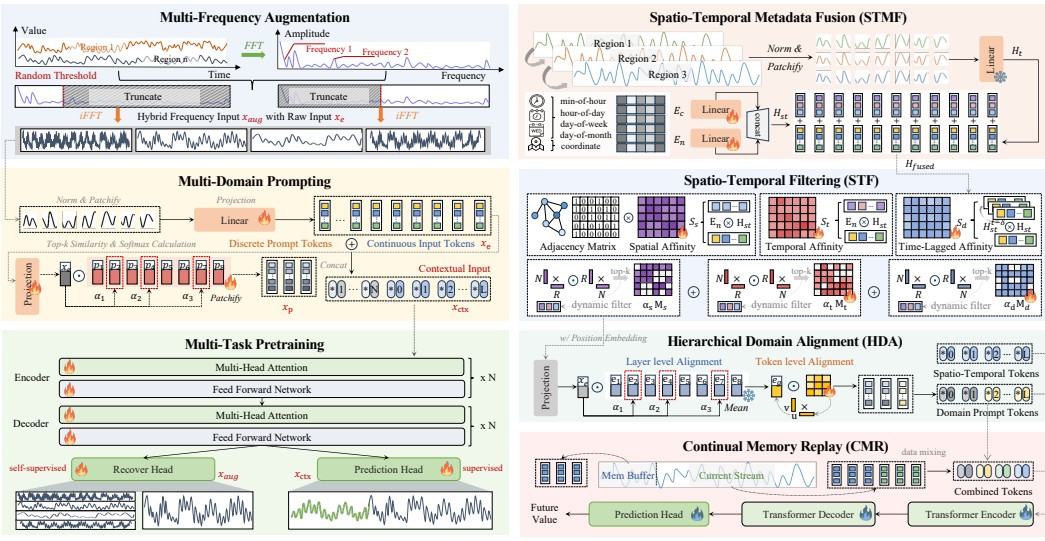

Figure 2: Overview of FactoST.

## 3 Methodology

Figure 2 presents the framework of FactoST for factorized STFM, consisting of two stages:

- Given cross-domain ST data $\mathbf{d} \in \mathbb{R}^{N \times L \times D}$ with $N$ nodes, $L$ time steps, and $D$-dimension, we apply a compact general-purpose temporal backbone $T : \mathbb{R}^{L \times D} \to \mathbb{R}^{F \times D}$ independently to each node's sequence $\mathbf{d}[i, :, :]$ to predict the future horizon $F$. This stage integrates multi-frequency augmentation, domain prompting, and multi-task to learn universal temporal patterns.

- A lightweight adapter $S$ is designed to rapidly adapt $T$ to specific ST domains. For downstream input $\mathbf{x}_{\text{in}} \in \mathbb{R}^{N \times L \times D}$, we reuse the pretrained backbone $T$ to extract node-wise temporal features $\mathbf{z} \in \mathbb{R}^{N \times L \times d}$, where $d$ denotes the model's hidden dimension. $S$—parameterized by $\Phi$ with $\|\Phi\| \ll \|T\|$—then refines these features by injecting ST metadata $\mathbf{m}$, pruning redundant ST interactions, aligning domain gaps, and performing strategic sample mixing for fine-tuning. The final output $\mathbf{y}_{\text{out}} = S(\mathbf{z}; \mathbf{m}) \in \mathbb{R}^{N \times F \times D}$ yields forecasts for the $F$-step future horizon.

### 3.1 Stage I: Universal Temporal Pretraining (UTP)

To distill transferable temporal dynamics across heterogeneous ST domains, we pretrain a spatially agnostic temporal backbone on node-wise time series using a Transformer encoder-decoder architecture. This stage deliberately omits any spatial graph structure, enabling scalable and domain-agnostic learning of universal temporal patterns such as periodicity, trends, and multi-scale fluctuations.

**Multi-Frequency Augmentation.** Temporal patterns often exhibit both long-term trends and short-term fluctuations. Drawing on the ideas from [63, 8, 60], we adopt a *frequency isolation* strategy to generate diverse temporal views that emphasize distinct spectral components. Given a raw input sequence $\mathbf{x} \in \mathbb{R}^{L \times D}$, we first apply the Fast Fourier Transform (FFT) to obtain its spectral representation $\mathbf{x}_f \in \mathbb{C}^F$, where $F = \lfloor L/2 \rfloor + 1$. We then stochastically isolate either low- or high-frequency bands by sampling $K_m$ random cutoff thresholds $\{\tau_i\}_{i=1}^{K_m}$ with $\tau_i \sim (0, F)$, and binary selectors $\{\mu_i\}_{i=1}^{K_m}$ with $\mu_i \sim \{0, 1\}$. For each pair $(\tau_i, \mu_i)$, we retain only the frequency components below $\tau_i$ if $\mu_i = 0$, or above $\tau_i$ if $\mu_i = 1$, effectively creating a *spectrally filtered* version of the signal. The filtered spectra are transformed back to the time domain via inverse FFT, yielding $K_m$ augmented views $\{\mathbf{x}_m^{(i)}\}_{i=1}^{K_m} \in \mathbb{R}^{L \times D}$. These views—along with the original input—are independently patched into non-overlapping segments of length $L'$ and projected into $d$-dimensional tokens. The resulting token sequences form a multi-view temporal tensor $\mathbf{x}_{\text{aug}} \in \mathbb{R}^{K_m \times N' \times L' \times d}$, where $N' = L/L'$, serving as input to the Transformer encoder. This design encourages the model to learn representations that are consistent across complementary frequency perspectives.

**Multi-Domain Prompting.** Drawing inspiration from codebooks in vision [43], we propose a *soft domain prompting* mechanism to encode cross-domain contextual cues. Specifically, we construct a learnable codebook $\mathbf{P} = \{p_1, \ldots, p_{K_p}\} \in \mathbb{R}^{K_p \times d}$, where each vector represents a prototypical

temporal context from a specific domain. Given an input $\mathbf{x}$, we first extract a compact embedding $\mathbf{x}_e \in \mathbb{R}^d$ via global pooling and a linear projection. We then compute its similarity with each prompt vector $p_j \in \mathbf{P}$ using negative squared Euclidean distance. Subsequently, we apply softmax normalization to derive attention weights, and obtain the final domain-specific prompt $\mathbf{x}_p \in \mathbb{R}^d$ via weighted combination, thereby fusing reusable knowledge across multiple domains. Finally, $\mathbf{x}_p$ is expanded into $N_p$ tokens and concatenated with the patched input to obtain $\mathbf{x}_{\text{ctx}} \in \mathbb{R}^{(N'+N_p)\times d}$.

$$s_j = -\|\mathbf{x}_e - p_j\|_2^2, \quad \alpha_j = \frac{\exp(s_j)}{\sum_{k=1}^{K_p}\exp(s_k)}, \quad \mathbf{x}_p = \sum_{j=1}^{K_p}\alpha_j p_j, \qquad j = 1,\dots,K_p. \tag{1}$$

where $s_j$ is the similarity score, $\alpha_j$ is the attention weight, and $\mathbf{x}_p$ is the domain context prompt.

**Multi-Task Pretraining.** To simultaneously capture universal temporal structures and enable effective cross-domain transfer, we jointly optimize two complementary objectives: a self-supervised task that enforces consistency across multi-frequency views of the input, and a supervised forecasting task that guides the model to learn predictive, domain-aware representations:

- *Self-supervised spectral consistency*: The model reconstructs the original time series from the multi-view augmented input $\mathbf{x}_{\text{aug}}$, ensuring learned representations preserve coherent information across complementary frequency bands. Specifically, $\mathbf{x}_{\text{aug}}$ is encoded by a Transformer to capture deep temporal interactions, then decoded via a Transformer decoder and a linear head to regenerate the original sequence, optimized with mean squared error (MSE) loss.

$$\mathcal{L}_{\text{spec}} = \big\|\mathbf{x} - \text{SpecHead}\big(\text{Decoder}(\text{Encoder}(\mathbf{x}_{\text{aug}}))\big)\big\|_2^2. \tag{2}$$

- *Supervised forecasting with prompt alignment*: The model forecasts future values from the domain-prompted input $\mathbf{x}_{\text{ctx}}$ to evaluate representation quality. $\mathbf{x}_{\text{ctx}}$ passes through the shared Transformer encoder-decoder, but the decoder output is detached before the prediction head—using the forecasting loss as a non-backpropagated supervisory signal. This ensures only the restoration task updates shared parameters, preventing negative transfer from task-specific biases.

$$\mathcal{L}_{\text{pred}} = \big\|\mathbf{y} - \text{PredHead}\big(\text{detach}\big(\text{Decoder}(\text{Encoder}(\mathbf{x}_{\text{ctx}}))\big)\big)\big\|_2^2 + \|\mathbf{x}_e - \mathbf{x}_p\|_2^2, \tag{3}$$

where $\mathbf{y}$ is the ground-truth. The first term minimizes supervised forecasting error, while the second enforces *prompt consistency* between the input embedding $\mathbf{x}_e$ and its soft domain prompt $\mathbf{x}_p$, inspired by the codebook alignment objective in VQ-VAE [54] to stabilize training and enhance representation fidelity. The pretraining objective combines both losses: $\mathcal{L} = \mathcal{L}_{\text{spec}} + \mathcal{L}_{\text{pred}}$. During subsequent ST fine-tuning, domain prompts are frozen, and only the Transformer layers and prediction head are updated using $\mathcal{L}_{\text{pred}}$, enabling efficient adaptation while preserving pretrained knowledge.

## 3.2 Stage II: Spatio-Temporal Adaptation

To adapt the pretrained temporal backbone to ST scenarios, we introduce four lightweight modules that incur minimal parameter overhead while effectively capturing ST dependencies.

**Spatio-Temporal Metadata Fusion (STMF).** This module injects ST context into the temporal backbone via learnable identifiers. Given ST input $\mathbf{X} \in \mathbb{R}^{N\times L\times D}$, we first get it temporal representations $H_t \in \mathbb{R}^{N\times N'\times d}$ via patch embedding layer, then we define: (1) node-specific spatial embeddings $\mathbf{E}_n \in \mathbb{R}^{N\times D_e}$; and (2) a calendar-aware temporal embedding bank $\{\mathbf{E}_c\}_{c\in\mathcal{S}}$, where each calendar type $c$ (e.g., minute-of-hour, hour-of-day, day-of-week, day-of-month, month-of-year) has an embedding table $\mathbf{E}_c \in \mathbb{R}^{K_c\times D_e}$, with $K_c$ the number of discrete bins for type $c$ (e.g., $60/24/7/31/12$). The active set $\mathcal{S}$ is chosen by sampling frequency (e.g., hourly: $\{\text{hour-of-day, day-of-week, day-of-month, month-of-year}\}$; minutely: add minute-of-hour).

For each node–patch pair $(i,\tau)$, we map the start timestamp to calendar bins via $\phi_c(\tau) \in \{1,\dots,K_c\}$ for each $c \in \mathcal{S}$, and form an ST identifier by concatenating the node embedding with the selected calendar embeddings as $\mathbf{h}_{\text{st}}(i,\tau) = \mathbf{W}_p\big[\mathbf{E}_n^i \,\big\|\, \big\|_{c\in\mathcal{S}}\mathbf{E}_c^{\phi_c(\tau)}\big] + \mathbf{b}_p$, where $\|$ denotes concatenation.

The identifiers' dimension are projected and expanded to obtain $\mathbf{H}_{\text{st}} \in \mathbb{R}^{N\times N'\times d}$, aligning with $H_t$. The encoded input representation and ST identifiers are then fused together by residual addition: $\mathbf{H}_{\text{fused}} = \mathbf{H}_t + \mathbf{H}_{\text{st}}$. This enables integration of ST context without retraining the temporal model.

**Spatio-Temporal Filtering (STF).** While STMF uses static ST identifiers, STF adapts to scenario-dependent cue relevance—e.g., local spatial context for incidents vs. global temporal patterns for rush hours—by dynamically reweighting spatial and temporal interactions via three learnable affinities. From $\mathbf{H}_{\text{st}} \in \mathbb{R}^{N \times N' \times d}$, we extract spatial ($\mathbf{E}_n$) and temporal ($\mathbf{E}_t$) embeddings and compute:

- **Spatial Affinity ($\mathbf{S}_s$):** Measures the compatibility between $\mathbf{H}_{\text{st}}$ and its spatial component $\mathbf{E}_n$ via dot-product: $\mathbf{S}_s = \langle \mathbf{H}_{\text{st}}, \mathbf{E}_n \rangle \in \mathbb{R}^{N \times N'}$, where higher values indicate stronger spatial relevance for each (node, patch) pair. This avoids rigid reliance on fixed spatial identifiers.
- **Temporal Affinity ($\mathbf{S}_t$):** Quantifies alignment between $\mathbf{H}_{\text{st}}$ and its temporal component $\mathbf{E}_t$: $\mathbf{S}_t = \langle \mathbf{H}_{\text{st}}, \mathbf{E}_t \rangle \in \mathbb{R}^{N \times N'}$, capturing dominant temporal patterns while filtering redundant noise.
- **Time-Lagged Affinity ($\mathbf{S}_d$):** Models asynchronous causal effects (e.g.,upstream nodes influencing downstream nodes with delay $\delta$). For lags $\delta = 1, \ldots, \Delta$, it aggregates historical neighbor states $\mathbf{H}_{\text{st}}^{(t-\delta)}$ and computes: $\mathbf{S}_d = \sum_{\delta=1}^{\Delta} \gamma^{(\delta)} \cdot \langle \mathbf{H}_{\text{st}}, \text{Agg}_\delta(\mathbf{H}_{\text{st}}^{(t-\delta)}) \rangle \in \mathbb{R}^{N \times N'}$, where $\gamma^{(\delta)}$ are learnable lag weights. Higher values reflect stronger delayed relevance.

To improve scalability, all affinity computations can be performed in a low-rank space. Specifically, we project both operands into $\mathbb{R}^r$ ($r \ll d$) via shared or separate learnable matrices, e.g., for spatial affinity: $\mathbf{S}_s = \langle \mathbf{H}_{\text{st}} \mathbf{W}_q^{(s)}, \mathbf{E}_n \mathbf{W}_k^{(s)} \rangle$, where $\mathbf{W}_q^{(s)}, \mathbf{W}_k^{(s)} \in \mathbb{R}^{d \times r}$. Analogous projections apply to $\mathbf{S}_t$ and $\mathbf{S}_d$. This reduces complexity from $\mathcal{O}(d)$ to $\mathcal{O}(r)$ per inner product while preserving semantic expressiveness. Top-$K$ sparsification may further prune weak interactions.

The three affinities are stacked as $\mathbf{S} = [\mathbf{S}_s, \mathbf{S}_t, \mathbf{S}_d] \in \mathbb{R}^{N \times N' \times 3}$, projected to dimension $d$ via $\mathbf{W}_{\text{att}} \in \mathbb{R}^{3 \times d}$, and normalized with a softmax (temperature $\tau_{\text{att}}$) to yield dynamic weights:

Finally, the refined output is obtained by aggregating the three affinity scores into $\mathbf{S} = [\mathbf{S}_s, \mathbf{S}_t, \mathbf{S}_d] \in \mathbb{R}^{N \times L \times 3}$, projecting to dimension $D$ via learnable matrix $\mathbf{W}_{\text{att}} \in \mathbb{R}^{3 \times d}$, normalizing with softmax (temperature $\tau_{\text{att}}$) to get dynamic weights $\mathbf{W}$, then modulating $\mathbf{H}_{\text{st}}$ with $\mathbf{W}$ and applying LayerNorm:

$$\mathbf{W} = \text{softmax}\left(\frac{\mathbf{S}\mathbf{W}_{\text{att}}}{\tau_{\text{att}}}\right), \quad \mathbf{H}_{\text{st}} = \text{LayerNorm}\left(\mathbf{H}_{\text{st}} \odot \mathbf{W}\right) \in \mathbb{R}^{N \times N' \times d}$$

This design enables adaptive integration of ST context without retraining the temporal backbone, effectively balancing spatial and temporal semantics while suppressing irrelevant cues.

**Hierarchical Domain Alignment (HDA).** To bridge the discrepancy across domains and facilitate effective transfer of domain adaptation knowledge, we propose a hierarchical alignment module using the pretrained domain prompts $\mathbf{p} \in \mathbb{R}^{K_p \times D}$, which operates at two levels:

1. **Layer-level alignment**: For an input embedding $\mathbf{x}_e \in \mathbb{R}^D$, we retrieve its $k$ nearest prompts in $\mathbf{p}$ based on negative Euclidean distance and compute a soft domain prototype via averaging:

$$\mathcal{K}(\mathbf{x}_e) = \underset{j \in [1, K_p]}{\text{Topk}} \left(-\|\mathbf{x}_e - \mathbf{p}_j\|_2\right), \quad \bar{\mathbf{p}}_k = \frac{1}{k} \sum_{j \in \mathcal{K}(\mathbf{x}_e)} \mathbf{p}_j \in \mathbb{R}^d. \quad (4)$$

2. **Token-level alignment**: To capture dataset-specific patterns beyond the pretrained prompt knowledge, we introduce a low-rank adaptation matrix $\mathbf{A} = \mathbf{u}\mathbf{v}^\top$, where $\mathbf{u} \in \mathbb{R}^{N_p}$ and $\mathbf{v} \in \mathbb{R}^D$. The domain-aware adjustment is then computed as $\mathbf{X}_r = \left(\mathbf{1}_{N_p} \bar{\mathbf{p}}_k^\top\right) \odot \mathbf{A} \in \mathbb{R}^{N_p \times d}$. Where $\mathbf{1}_{N_p}$ is a column vector of ones and $\odot$ denotes element-wise multiplication. The final representation fuses the layer-level prototype and token-level refinement for enhanced cross-domain generalization.

**Continual Memory Replay (CMR).** To mitigate knowledge forgetting during few-shot adaptation, we implement dynamic data mixing, combining current data and historical data. First, we establish a memory buffer, given training sequences $\{\mathbf{X}_t\}_{t=1}^T$ of length $T$, we partition the dataset into:

$$\mathcal{M} = \{\mathbf{X}_t\}_{t=1}^{T_m} \text{ (memory buffer)}, \quad \mathcal{C} = \{\mathbf{X}_t\}_{t=T_m+1}^T \text{ (current stream)}, \quad (5)$$

with $T_m = \lfloor \text{memory\_size} \cdot T \rfloor$ (default: 0.2), where the memory buffer $\mathcal{M}$ preserves critical temporal patterns from initial learning to ensure stability under domain shift.

Each mini-batch $\mathcal{B}$ is constructed by strategically mixing samples from both current stream and memory buffer: $\mathcal{B} = \{\mathbf{X}_i\}_{i \in \mathcal{I}_c \setminus \mathcal{R}} \cup \{\mathbf{X}_j\}_{j \in \mathcal{I}_m[:|\mathcal{R}|]}$, where $\mathcal{I}_c$ and $\mathcal{I}_m$ denote shuffled indices of $\mathcal{C}$ and $\mathcal{M}$, respectively, and $\mathcal{R} \subset \mathcal{I}_c$ has size $\lfloor r \cdot |\mathcal{B}| \rfloor$, with $r = 0.3$ by default. This mechanism effectively preserves implicit historical knowledge during domain adaptation.

# 4 Experiments

In our experiments, we aim to address the following research questions (RQ):

- **RQ1**: Can FactoST outperform prior approaches (including STGNNs, STFMs and other existing models) under few-shot and zero-shot scenarios? ⇒ **Sec. 4.1 & Sec. 4.2**.
- **RQ2**: Which model component is critical to the final performance? ⇒ **Sec. 4.3.1**.
- **RQ3**: How is the data and computation efficiency of FactoST? ⇒ **Sec. 4.3.2 & Sec. 4.3.3**.
- **RQ4**: Can we provide interpretability of the domain adaptation process in FactoST? ⇒ **Sec. 4.3.4**.
- **RQ5**: Is the STA module architecture-agnostic, or limited to GNN-based backbones? ⇒ **Sec. 4.3.5**.

**Datasets.** We pretrain the temporal backbone on diverse ST datasets using Monash [16], covering six domains (energy, nature, health, transport, web, economics) with 130M observations across multiple spatial nodes and sampling frequencies from 4 seconds to daily. During pretraining, we extract and process univariate time series per node independently to prevent data leakage. For evaluation, we use eight established ST benchmarks—traffic flow (PEMS03/04/07/08), speed (PEMS-BAY, METR-LA), energy (Electricity), temperature (ETTh2), and climate (Weather)—which vary widely in spatial scale (21–883 nodes), temporal resolution (5 min–1 h), and sequence length (17k–52k steps), enabling a comprehensive assessment of cross-domain and multi-scale generalization (see A.1.1 for details).

**Baselines.** We compare FactoST with 12 competitive models across four categories: 1) STFMs: OpenCity [33], UniST [69]; 2) TSFMs: TimesFM [10], Moirai [62]; 3) ST expert models: BigST [20], STAEformer [37], STID [48], D2STGNN [49]; 4) Time series expert models: TimeMixer [59], PatchTST [42], DLinear [71], Informer [79]. Consistent with previous works [69], we adopted Mean Absolute Error (MAE) and Root Mean Square Error (RMSE) as evaluation metrics. More in A.1.

## 4.1 Few-shot Prediction

**Setting.** We evaluate few-shot adaptation using only 10% of labeled training data under two forecasting horizons: short-term (12 → 12) and long-term (96 → 96), following standard protocols [49, 48].

**Results.** As shown in Tables 2 and 3, FactoST consistently outperforms all baselines across both short- and long-term horizons under few-shot adaptation. In the short-term setting (12 → 12), FactoST improves MAE over STFMs—OpenCity and UniST—by 31.4% and 47.2%, respectively, where UniST suffers from its rigid grid-based design and OpenCity incurs high graph-learning overhead. Against TSFMs (TimesFM, Moirai), we observe average gains of 18.8%, demonstrating that our lightweight ST adaptation effectively enriches universal temporal representations. Notably, FactoST remains competitive with specialized ST expert models despite avoiding domain-specific

Table 2: Few-shot short-term forecasting (12→12) results on 10% training data across multiple ST datasets. Lower values indicate better performance. **Red**: the best. Blue: the second best.

| Method Type | | Foundation Model | | | | | Expert Model | | | | | | | |
| --- | --- | --- | --- | --- | --- | --- | --- | --- | --- | --- | --- | --- | --- | --- |
| | | *Spatio Temporal* | | | *Time Series* | | *Spatio Temporal* | | | | *Time Series* | | | |
| Method | | FactoST | OpenCity | UniST | TimesFM | Moirai | BigST | STAEformer | STID | D2STGNN | TimeMixer | PatchTST | DLinear | Informer |
| PEMS-03 | MAE | **17.54** | 17.90 | 40.39 | 21.99 | 21.40 | 18.41 | 30.79 | 22.93 | 18.55 | 21.41 | 21.97 | 21.94 | 23.24 |
| | RMSE | **28.10** | 28.80 | 53.44 | 35.31 | 32.38 | 28.45 | 47.67 | 34.10 | 29.21 | 33.57 | 35.59 | 35.30 | 37.98 |
| PEMS-04 | MAE | **23.93** | 24.78 | 42.76 | 27.84 | 33.73 | 23.97 | 48.23 | 26.72 | 24.86 | 27.37 | 28.11 | 28.37 | 29.81 |
| | RMSE | 37.44 | 40.41 | 59.07 | 43.15 | 54.09 | **36.88** | 68.46 | 40.31 | 38.43 | 42.16 | 44.13 | 44.57 | 45.59 |
| PEMS-07 | MAE | 26.48 | 44.43 | 40.77 | 32.61 | 35.69 | 25.72 | 33.50 | 31.46 | **25.51** | 30.31 | 31.19 | 31.89 | 37.55 |
| | RMSE | 41.92 | 65.47 | 54.86 | 50.20 | 51.36 | **39.72** | 51.43 | 46.72 | 39.81 | 46.36 | 48.91 | 49.65 | 62.55 |
| PEMS-08 | MAE | **18.94** | 32.16 | 35.70 | 22.06 | 38.01 | 19.40 | 36.15 | 23.17 | 19.55 | 22.05 | 22.42 | 23.10 | 31.69 |
| | RMSE | **29.59** | 48.47 | 46.74 | 33.87 | 53.05 | 29.96 | 51.05 | 34.09 | 30.51 | 34.09 | 35.64 | 36.35 | 51.53 |
| PEMS-Bay | MAE | 1.96 | 2.77 | 5.14 | 2.25 | 2.26 | **1.91** | 2.01 | 2.00 | 1.99 | 2.11 | 2.15 | 2.21 | 2.96 |
| | RMSE | 4.51 | 6.08 | 8.28 | 5.49 | 5.49 | **4.26** | 4.62 | 4.57 | 4.72 | 4.93 | 5.23 | 5.20 | 6.23 |
| METR-LA | MAE | 4.77 | 4.18 | 8.79 | 5.56 | 4.95 | 3.72 | 4.61 | 4.00 | 4.00 | 4.23 | 4.34 | 4.57 | 4.93 |
| | RMSE | 9.88 | 8.33 | 14.34 | 12.87 | 12.75 | 7.19 | 8.91 | 8.20 | 8.03 | 9.20 | 9.75 | 9.82 | 9.20 |
| ETTh2 | MAE | 0.272 | 0.513 | 0.425 | 0.284 | **0.135** | 0.740 | 1.208 | 0.756 | 0.916 | 0.803 | 0.721 | 1.885 | 2.125 |
| | RMSE | 0.424 | 0.710 | 0.545 | 0.410 | **0.307** | 1.214 | 1.673 | 1.224 | 1.433 | 1.228 | 1.211 | 2.946 | 2.898 |
| Electricity | MAE | **0.374** | 0.412 | 0.565 | 0.529 | 0.837 | 0.638 | 0.858 | 0.575 | 0.686 | 0.767 | 0.840 | 1.282 | 1.598 |
| | RMSE | **0.545** | 1.740 | 3.276 | 0.801 | 1.036 | 4.545 | 8.289 | 1.085 | 4.535 | 4.324 | 5.097 | 8.837 | 15.649 |
| Weather | MAE | **0.087** | 0.414 | 0.239 | 0.138 | 0.184 | 0.375 | 0.575 | 0.330 | 0.587 | 0.311 | 0.296 | 0.383 | 0.958 |
| | RMSE | **0.276** | 0.660 | 0.381 | 0.323 | 0.432 | 0.951 | 1.085 | 0.920 | 1.269 | 0.967 | 1.074 | 1.046 | 1.783 |
| Count | 1st | **9** | 0 | 0 | 2 | 2 | 4 | 0 | 1 | 3 | 0 | 0 | 0 | 0 |
| | 2nd | 4 | 4 | 0 | 4 | 0 | 4 | 0 | 1 | 1 | 0 | 0 | 0 | 0 |

Table 3: Few-shot long-term forecasting (96 → 96) results on 10% training data across multiple spatio-temporal datasets. Lower values indicate better performance. **Red**: the best, Blue: the second

| | | Foundation Model | | | | | Expert Model | | | | | | | |
| | | Spatio Temporal | | | Time Series | | Spatio Temporal | | | | Time Series | | | |
| Method Type | Method | FactoST | OpenCity | UniST | TimesFM | Moirai | BigST | STAEformer | STID | D2STGNN | TimeMixer | PatchTST | DLinear | Informer |
| PEMS-03 MAE | | **28.57** | 34.21 | 67.70 | 38.47 | 51.40 | 51.87 | 77.42 | 45.45 | OOM | 47.86 | 61.22 | 76.41 | 46.27 |
| RMSE | | **46.78** | 54.82 | 94.00 | 59.77 | 79.47 | 75.56 | 115.67 | 65.35 | OOM | 71.52 | 100.33 | 113.63 | 69.41 |
| PEMS-04 MAE | | **42.04** | 67.24 | 85.14 | 64.43 | 81.30 | 52.37 | 64.12 | 78.13 | OOM | 58.44 | 70.71 | 85.61 | 54.26 |
| RMSE | | **64.89** | 112.20 | 112.11 | 93.44 | 116.26 | 80.23 | 91.95 | 111.12 | OOM | 86.67 | 104.00 | 125.44 | 83.42 |
| PEMS-07 MAE | | **45.60** | 50.70 | 101.20 | 157.10 | 134.46 | 54.92 | 61.45 | 71.32 | OOM | 67.75 | 80.09 | 106.68 | 52.82 |
| RMSE | | **72.47** | 78.36 | 134.98 | 208.36 | 200.30 | 82.12 | 91.06 | 106.95 | OOM | 100.45 | 118.54 | 147.43 | 81.78 |
| PEMS-08 MAE | | **35.69** | 49.47 | 73.81 | 89.93 | 68.73 | 58.68 | 68.45 | 75.87 | OOM | 45.10 | 57.31 | 76.77 | 44.25 |
| RMSE | | **56.15** | 82.07 | 96.45 | 125.27 | 97.89 | 86.76 | 96.14 | 103.15 | OOM | 65.53 | 87.33 | 109.15 | 68.43 |
| PEMS-Bay MAE | | 2.96 | 7.40 | 5.17 | 5.18 | 5.78 | **2.93** | 3.28 | 3.10 | OOM | 4.11 | 4.32 | 4.62 | 3.27 |
| RMSE | | 6.21 | 12.38 | 8.27 | 9.97 | 10.97 | **6.20** | 6.65 | 6.63 | OOM | 8.94 | 9.22 | 9.52 | 6.81 |
| METR-LA MAE | | 6.93 | 9.71 | 13.16 | 14.23 | 12.17 | 6.69 | 6.15 | **5.94** | OOM | 7.00 | 7.20 | 7.65 | 6.38 |
| RMSE | | 13.07 | 13.62 | 19.96 | 22.56 | 22.39 | 12.22 | **11.56** | 11.91 | OOM | 13.33 | 14.17 | 13.42 | 12.29 |
| ETTh2 MAE | | 0.358 | 0.751 | 0.488 | 0.365 | **0.325** | 1.164 | 1.295 | 1.066 | OOM | 1.013 | 0.943 | 1.069 | 2.960 |
| RMSE | | 0.561 | 1.040 | 0.622 | 0.541 | **0.465** | 1.797 | 1.918 | 1.751 | OOM | 1.646 | 1.609 | 1.781 | 3.783 |
| Electricity MAE | | **0.265** | 0.303 | 0.494 | 0.305 | 0.312 | 0.481 | 0.733 | 0.440 | OOM | 0.459 | 0.442 | 0.558 | 1.693 |
| RMSE | | **0.409** | 1.240 | 2.512 | 0.465 | 0.484 | 2.843 | 6.562 | 2.755 | OOM | 2.833 | 2.747 | 3.262 | 16.536 |
| Weather MAE | | **0.226** | 0.653 | 0.348 | 0.270 | 0.262 | 0.892 | 1.171 | 0.740 | OOM | 0.720 | 0.708 | 0.731 | 2.249 |
| RMSE | | **0.426** | 3.730 | 0.491 | 0.484 | 0.465 | 1.522 | 1.804 | 1.446 | OOM | 1.401 | 1.409 | 1.424 | 3.403 |
| Count 1st | | **12** | 0 | 0 | 0 | 2 | 2 | 1 | 1 | 0 | 0 | 0 | 0 | 0 |
| 2nd | | 3 | 5 | 0 | 2 | 2 | 0 | 1 | 1 | 0 | 0 | 0 | 0 | 4 |

spatial modules like graph networks, and substantially outperforms time series experts that ignore spatial structure—especially on spatially correlated tasks. The advantage further widens in the long-term setting (96 → 96): FactoST achieves average MAE reductions of 40.5% over STFMs, 33.8% over TSFMs, and remarkably 44.1% over ST experts, even as models like D2STGNN fail due to out-of-memory errors under long-range forecasting. This underscores the scalability and efficiency of our factorized design: by decoupling universal temporal pretraining from plug-and-play ST adaptation, FactoST captures long-range dependencies without end-to-end joint ST pretraining or heavy spatial inductive biases—making it particularly effective in low-data, long-horizon scenarios.

Table 4: Zero-shot performance comparison of foundation models on short-term (12 → 12) and long-term (96 → 96) forecasting across ST datasets. * indicates the dataset was seen in pretraining; results marked with * are **excluded** when determining **Red** (best) and Blue (second-best).

| Dataset | Horizon | FactoST | | OpenCity | | UniST | | TimesFM | | Moirai | |
| | | MAE | RMSE | MAE | RMSE | MAE | RMSE | MAE | RMSE | MAE | RMSE |
| PEMS-Bay | Short | **2.02** | **4.59** | 3.23* | 6.91* | 14.89 | 16.92 | 6.59 | 14.99 | 1.97* | 4.69* |
| | Long | **6.12** | **10.47** | 6.92* | 11.79* | 9.29 | 12.72 | 14.16 | 24.01 | 6.51* | 11.70* |
| METR-LA | Short | **4.85** | **10.12** | 4.30* | 8.37* | 24.33 | 29.31 | 6.59 | 14.99 | 5.55 | 13.79 |
| | Long | **12.68** | **18.82** | 10.85* | 17.60* | 25.88 | 30.19 | 14.16 | 24.01 | 12.87 | 22.19 |
| PEMS-03 | Short | 30.12 | **46.92** | 30.37 | 47.49 | 101.87 | 129.94 | **29.71** | 49.21 | 28.23* | 46.36* |
| | Long | 113.62 | 144.80 | 125.18 | 159.23 | **102.87** | **137.90** | 108.59 | 152.74 | 74.85* | 110.78* |
| PEMS-04 | Short | 38.65 | 56.59 | 39.34* | 58.50* | 67.91 | 87.63 | **35.00** | **53.27** | 34.65* | 52.13* |
| | Long | 143.11 | 175.99 | 153.18* | 188.95* | **115.76** | **153.58** | 127.39 | 171.13 | 105.95* | 141.40* |
| PEMS-07 | Short | 45.47 | 66.68 | 45.18 | 67.15 | 104.93 | 133.65 | **42.10** | **64.59** | 35.64* | 50.25* |
| | Long | 156.07 | 190.59 | 172.20 | 211.25 | **144.67** | **184.62** | 151.99 | 205.18 | 125.91* | 169.43* |
| PEMS-08 | Short | 32.14 | 47.27 | 32.45* | 48.42* | 73.46 | 93.14 | **29.68** | **45.18** | 38.23* | 53.12* |
| | Long | 121.43 | 151.59 | 128.48* | 161.75* | 104.77 | 136.21 | **92.72** | **126.10** | 119.10* | 151.83* |
| Count | 1st | 4 | 5 | 0 | 0 | 3 | 3 | 5 | 4 | 0 | 0 |
| | 2nd | 3 | 5 | 0 | 1 | 2 | 2 | 4 | 2 | 2 | 2 |

## 4.2 Zero-shot Prediction

**Setting.** We evaluate zero-shot forecasting for both short- and long-term scenarios without fine-tuning.

**Results.** As shown in Table 4, explicit joint ST pretraining does not improve zero-shot generalization. In fact, models without dedicated spatial modeling—such as FactoST (UTP without STA) and TimesFM—consistently outperform specialized STFMs like OpenCity and UniST. This confirms our core insight: *spatial structures are highly domain-specific and hinder transfer when baked into pretraining*. UniST's unstable performance—reasonable on long-term but catastrophic on short-term

tasks—and its need for retraining under topology changes further expose the rigidity of fixed spatial priors. Even in-domain, OpenCity and Moirai (a multivariate TSFM) underperform, underscoring that strong zero-shot capability stems from temporal, not spatial, modeling. Remarkably, FactoST achieves the most top-2 rankings (4 first, 5 second) among all foundation models on unseen domains, despite using only 13M pretraining points—orders of magnitude fewer than Moirai (27B) or TimesFM (100B) (Table 6). Nonetheless, errors remain high in long-term scenarios, highlighting the intrinsic challenge of zero-shot ST prediction. To address this, we apply Test-Time Computing [9], which reduces zero-shot MAE and RMSE by 7.72% and 8.79% on average; see Appendix A.4.4 for details.

## 4.3 Model Analysis

**Ablation Studies.** We evaluate each component's contribution via ablation on PEMS-03 short-term forecasting (Figure 3). Removing CMR causes the largest drop (↑35.15% MAE, ↑32.85% RMSE), underscoring its role in preserving knowledge during few-shot adaptation. Disabling HDA or STMF degrades MAE by 22%, confirming their importance for domain alignment and metadata fusion. Among STF variants, omitting the spatial affinity matrix yields the sharpest decline (↑29.01% MAE, ↑33.70% RMSE), highlighting its efficacy in dynamic interaction modeling; temporal and time-delay matrices also contribute, albeit more moderately. We also find that the spectral consistency loss $\mathcal{L}_{\text{spec}}$ in pretraining provides nontrivial gains, more details are in Appendix A.4.3.

**Scaling Analysis.** We evaluate FactoST on PEMS-03 across downstream fine-tuning proportions from zero-shot to full-shot for both short- and long-term forecasting (Figure 4). Remarkably, FactoST achieves rapid performance gains with minimal data: in the short-term setting, MAE drops from 25.96 (1% data) to 17.54 (10% data)—already approaching full-shot performance (16.59). In the long-term setting, MAE plummets from 123.57 (zero-shot) to 28.57 with just 10% of the training data, and further improves to 25.85 under full supervision. This sharp improvement highlights FactoST 's exceptional data efficiency and fast adaptation capability. We also analyze scaling with respect to model size and pretraining data volume, observing diminishing returns beyond moderate capacity but consistent gains with more pretraining data; more details are in Appendix A.4.1 and A.4.2.

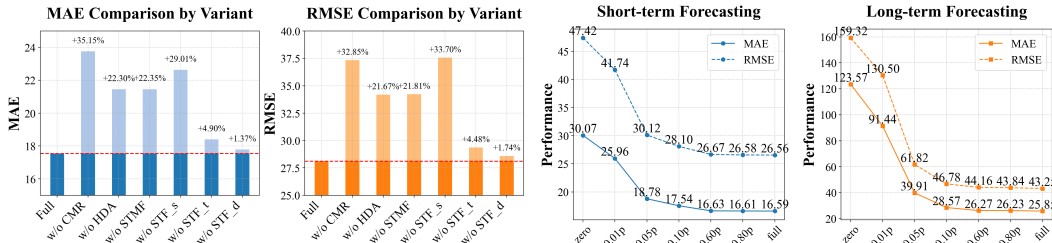

Figure 3: Ablation studies of various components.          Figure 4: Data scaling analysis.

**Efficiency Analysis.** As shown in Figure 5, FactoST achieves a strong MAE of 17.86 on PEMS-03 under the 10% few-shot setting with only 1.3M parameters and 8.1s inference time—outperforming nearly all baselines in both accuracy and efficiency. In contrast, joint ST models like OpenCity (1.67M params, 25.3s) incur high computational overhead due to end-to-end graph learning, while large-scale TSFMs such as Moirai (91.4M params, 22.2s) ignore spatial structure entirely. Even though models like D2STGNN attain slightly better accuracy, they suffer from severe latency (54.5s). By decoupling universal temporal pretraining from lightweight, plug-and-play spatial adaptation, FactoST achieves an exceptional *accuracy–efficiency trade-off*, making it highly suitable for real-world deployment.

**Domain Adaptation Analysis.** Figure 6 visualizes ST token embeddings before and after Hierarchical Domain Adaptation (HDA) via t-SNE: original tokens (blue), adapted tokens (orange), and learned domain prompts (circled clusters). The adapted embeddings shift clearly toward their corresponding prompt clusters, demonstrating that HDA effectively steers generic temporal representations toward domain-specific semantics. Crucially, the global structure of the embedding space is preserved—indicating that *adaptation is targeted and non-destructive*. This provides empirical evidence that FactoST successfully bridges universal knowledge from pretraining with task-specific spatio-temporal context, enabling effective cross-domain transfer without catastrophic forgetting.

**Architecture Generality of STA.** The STA module is *architecture-agnostic*, operating solely on feature embeddings without relying on GNN-specific inductive biases. To verify this, we integrate STA into PatchTST—a non-GNN, Transformer-based time series model—and observe consistent few-shot improvements across all datasets ( Figure 7). This confirms STA's ability to inject ST context into diverse backbones. FactoST remains superior, benefiting from large-scale pretraining; this highlights that *temporal knowledge learned from diverse domains generalizes effectively*, and when combined with lightweight spatial adaptation, enables strong cross-domain performance.

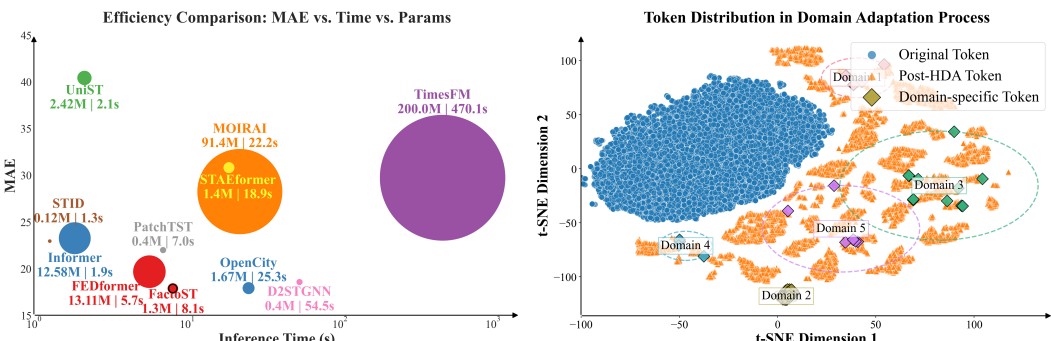

Figure 5: Efficiency comparison with baselines.

Figure 6: Domain adaptation visualization.

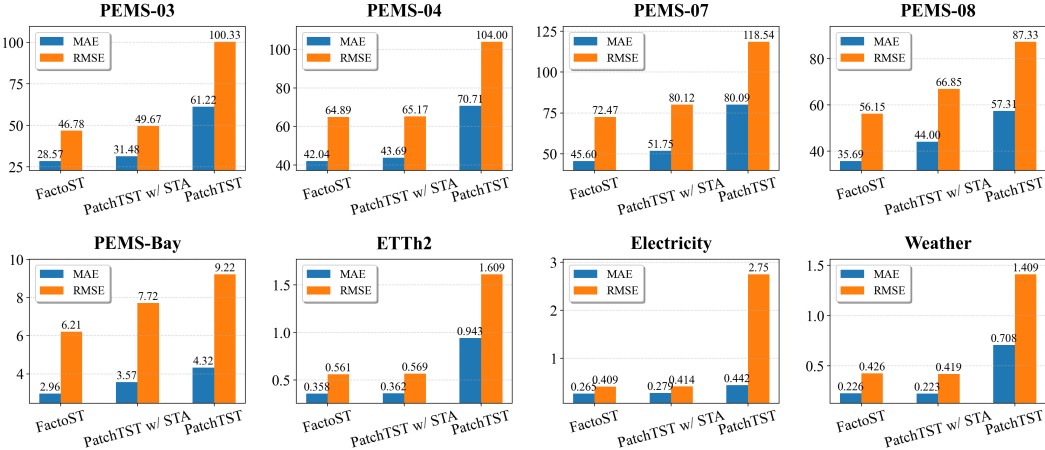

Figure 7: Few-shot long-term forecasting comparison of FactoST, PatchTST, and PatchTST with STA. Results confirm that STA is architecture-agnostic and consistently improves non-GNN backbones.

## 5 Conclusion and Future Work

We introduce FactoST, a two-stage spatio-temporal foundation model (STFM) that decouples universal temporal pretraining from lightweight spatio-temporal adaptation. This factorized design avoids the computational cost and poor generalization of joint ST pretraining in existing STFMs. Empirically, FactoST outperforms current STFMs in few-shot settings—reducing MAE by up to 46.4% over UniST—while using 46.2% fewer parameters and achieving 68% faster inference than OpenCity. Notably, it matches or surpasses domain-specific models without architectural customization, demonstrating the power of factorized pretraining as a scalable path toward universal STFMs.

We identify two key directions for future work. First, *spatial generalization remains a bottleneck*: limited zero-shot performance across STFMs—including our temporal-only backbone—suggests that rigid spatial priors impede cross-domain transfer, calling for more adaptive, semantics-aware representations. Second, the *fine-tuning protocol can be improved*: uniform parameter updates underutilize pretrained knowledge; parameter-efficient strategies (e.g., prompt tuning or selective retraining) could enhance transfer efficiency and mitigate catastrophic forgetting.

## Acknowledgments and Disclosure of Funding

This work was mainly supported by Huawei (Grant No. TC20241023027); the National Natural Science Foundation of China (Grant No. 62402414); the Guangdong Basic and Applied Basic Research Foundation (Grant No. 2025A1515011994); the Guangzhou Municipal Science and Technology Project (Grant No. 2023A03J0011); and the Guangzhou Industrial Information and Intelligent Key Laboratory Project (Grant No. 2024A03J0628).

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

## A  Technical Appendices

### A.1  Implementation Details

This section provides a comprehensive overview of the implementation setup, including datasets, evaluation metrics, hyperparameters, implementation details, and training configurations.

#### A.1.1  Datasets

**Pretraining Datasets:**    As shown in Table 5, we use cross-domain large-scale ST datasets, including energy, nature, transportation, and web, to train our temporal backbone. From these datasets, we extract node-wise temporal sequences across different locations over time. These sequences span multiple frequencies (from seconds to daily) and exhibit various temporal patterns, ensuring that the learned representations are robust and transferable across different forecasting tasks. We also add a data volume comparison in Table 6, where FactoST is much lower than other foundation models.

Table 5: List of pretraining spatio-temporal datasets.

| Dataset | Domain | Frequency | # Time Points | Source |
|---|---|---|---|---|
| Aus. Electricity Demand | Energy | Half Hourly | 1155264 | Monash[16] |
| Wind | Energy | 4 Seconds | 7397147 | Monash[16] |
| Wind Farms | Energy | Minutely | 172178060 | Monash[16] |
| Solar | Energy | 10 Minutes | 7200720 | Monash[16] |
| Solar Power | Energy | 4 Seconds | 7397222 | Monash[16] |
| London Smart Meters | Energy | Half Hourly | 166527216 | Monash[16] |
| Temperature Rain | Nature | Daily | 23252200 | Monash[16] |
| Saugeen River Flow | Nature | Daily | 23741 | Monash[16] |
| Sunspot | Nature | Daily | 73924 | Monash[16] |
| Weather | Nature | Daily | 43032000 | Monash[16] |
| KDD Cup 2018 | Nature | Daily | 2942364 | Monash[16] |
| US Births | Nature | Daily | 7305 | Monash[16] |
| Pedestrian_Counts | Transport | Hourly | 3132346 | Monash[16] |
| Web Traffic | Web | Daily | 116485589 | Monash[16] |
| Bitcoin | Economic | Daily | 75364 | Monash[16] |

Table 6: Pretraining corpora and scale of STFMs and TSFMs used in this study.

| Model | Pretraining Corpus | Scale |
|---|---|---|
| FactoST | Monash (5 domains, 15 datasets) | 13M time points |
| OpenCity | 21 heterogeneous traffic datasets (10,110 regions) | 151.1M observations |
| UniST | 21 multi-source grid datasets | — |
| TimesFM | Large-scale real-world and synthetic time series | 100B time points |
| Moirai | LOTSA dataset | 27B time points |

**Evaluation Datasets:**    For downstream ST forecasting, we select real-world benchmarks covering traffic flow, speed, electricity consumption, and meteorological data, as detailed in Table 7. These datasets exhibit substantial heterogeneity in spatial granularity (e.g., city regions vs. sensors), temporal resolution (e.g., 5-minute vs. hourly intervals), and prediction targets (e.g., speed vs. volume), posing a challenging testbed for cross-task generalization. This diversity enables a rigorous evaluation of FactoST 's adaptability under both short-term and long-term forecasting settings.

Table 7: List of evaluation spatio-temporal datasets.

| Dataset | Category | # Features | Sample rate | Time span (Y/M/D) | # Time Points |
|---------|----------|-----------|-------------|-------------------|---------------|
| PEMS03 | Traffic flow | 358 | 5 minutes | 2018/09/01 − 2018/11/30 | 26208 |
| PEMS04 | Traffic flow | 307 | 5 minutes | 2018/01/01 − 2018/02/28 | 16992 |
| PEMS07 | Traffic flow | 883 | 5 minutes | 2017/05/01 − 2017/08/31 | 28224 |
| PEMS08 | Traffic flow | 170 | 5 minutes | 2016/07/01 − 2016/08/31 | 17856 |
| PEMS-BAY | Traffic speed | 325 | 5 minutes | 2017/01/01 − 2017/06/30 | 52116 |
| METR-LA | Traffic speed | 207 | 5 minutes | 2012/03/01 − 2012/06/27 | 34272 |
| ETTh2 | Transformer temperature | 7 | 1 hour | 2016/07/01 − 2018/06/26 | 14400 |
| Electricity | Electricity consumption | 321 | 1 hour | 2012/01/01 − 2014/12/31 | 26304 |
| Weather | Meteorological data | 21 | 10 minutes | 2020/01/01 − 2021/01/01 | 52696 |

### A.1.2 Model Architecture and Hyperparameters

We implement FactoST using PyTorch, and all experiments are conducted on high-performance GPU servers. The architecture consists of three encoder layers and three decoder layers, with 16 attention heads and a latent dimension $d = 128$. Input sequences are processed using a patching mechanism with a patch size of 12, and the dropout rate is set to 0.2 to prevent overfitting. The feed-forward network within each Transformer layer has a hidden dimension of 512.

**Pretraining.** During pretraining, we use the Adam optimizer with an initial learning rate of $5 \times 10^{-4}$, and apply StepLR to decay the learning rate by a fixed factor every few epochs, improving convergence. The model is equipped with $N_p = 8$ domain prompt learning vectors, each of dimension 128, and in supervised prediction tasks, both the input length and target forecasting horizon are fixed at 96 (The length can be set to any value, which is the maximum supported step length, here we set 96 for downstream comparison). For spectral consistency modeling, the number of augmented patches is set to $K_{\mathbf{f}} = 4$. Pretraining is performed with a large batch size of 16,384 to ensure stable optimization.

**Fine-tuning.** During fine-tuning, we adopt a learning rate of $1 \times 10^{-3}$. The lookback window is set to 12 (short-term) or 96 (long-term), with matching prediction horizons. The number of domain prompt tokens ($N_p = 3$) and patching configuration remain unchanged from pretraining. A top-$k$ selection ($k = 3$) is applied during domain prompt matching to enhance generalization. Additional configuration includes memory replacement ratio of 0.3; memory size of 0.2 relative to total capacity; spatio-temporal identifier embedding dimension of 32; maximum delay step $\Delta = 3$.

### A.2 Evaluation Metrics

We use commonly used regression metrics, Mean Absolute Error (MAE) and Root Mean Squared Error (RMSE), to measure the prediction performance. Suppose $\mathbf{Y} = Y_1, ..., Y_M$ are ground truth for real spatio-temporal data, $\hat{\mathbf{Y}} = \hat{Y}_1, ..., \hat{Y}_N$ are the predicted values by the model, and $N$ is the number of total testing samples, These two metrics can be formulated as follows:

$$\text{RMSE}(\mathbf{Y}, \hat{\mathbf{Y}}) = \sqrt{\frac{1}{N} \sum_{i}^{N} \left( Y_i - \hat{Y}_i \right)^2}, \text{MAE}(\mathbf{Y}, \hat{\mathbf{Y}}) = \frac{1}{N} \sum_{i}^{N} \left| Y_i - \hat{Y}_i \right|, \qquad (6)$$

### A.3 Baselines and Implementation

All baseline models are evaluated within a unified framework to ensure fair comparisons. All models are assessed using standardized metrics (MAE and RMSE). Hyperparameters are either set to default values reported in the original papers or tuned via grid search on the validation set. Below, we detail the implementation strategies for each category of baselines.

### A.3.1 Time Series (TS) Expert Models

The following TS expert models are implemented using the `BasicTS` framework (`https://github.com/GestaltCogTeam/BasicTS`) to ensure consistency in preprocessing, training, and evaluation:

- **TimeMixer** [59]: A MLP-based model with past-decomposable mixing and future multi-predictor mixing, enabling multiscale information fusion from both microscopic and macroscopic views.

- **PatchTST** [42]: A Transformer-based model that treats time series as a sequence of patches, enabling effective long-term forecasting by capturing local and global patterns.

- **DLinear** [71]: A simple yet effective linear model that decomposes time series into trend and residual components, followed by independent modeling of each component for accurate forecasting.

- **Informer** [79]: An efficient Transformer variant with self-attention compression and generative decoder design, tailored for long sequence time series forecasting.

### A.3.2 Spatio-Temporal (ST) Expert Models

The following ST expert models are also integrated into the `BasicTS` framework, Graph structures follow original designs, utilizing distance-based or learned adjacency matrices where applicable.

- **BigST** [20]: Proposes a linear STGNN, first extracts long sequence input into a low representation, then uses a global GCN to capture spatial features, effective for large sensor node scenarios.

- **STAEformer** [37]: Utilizes spatial-temporal adaptive embeddings to enhance the representation learning capability of Transformers for traffic forecasting tasks.

- **STID** [48]: Introduces spatial-temporal identity vectors into the Transformer architecture to capture node-specific temporal dynamics and spatial dependencies.

- **D2STGNN** [49]: Decouples spatial and temporal dependencies using separate graph convolution and recurrent modules for improved modeling of complex spatio-temporal interactions.

### A.3.3 Time Series Foundation Models (TSFMs)

For TSFMs, we adapt the official implementations to align with our benchmarking protocol:

- **TimesFM** [10]: A large-scale pretrained decoder-only time series foundation model developed by Google Research, capable of high-accuracy univariate forecasting across diverse domains and frequencies. The implementation is obtained from the official repository (`https://github.com/google-research/timesfm`), and we fine-tune them using context lengths and forecast horizons consistent with our experimental setup. The checkpoint of the model we use comes from `https://huggingface.co/google/timesfm-1.0-200m`.

- **Moirai** [62]: A large-scale pretrained encoder-only time series foundation model developed by Salesforce AI Research, designed to deliver universal forecasting capabilities across diverse domains, frequencies, and variable types. The implementation is obtained from the official repository `https://github.com/SalesforceAIResearch/uni2ts`. The checkpoint of the model we use comes from `https://huggingface.co/Salesforce/moirai-1.0-R-base`.

### A.3.4 Spatio-Temporal Foundation Models (STFMs)

We evaluate two recent STFMs:

- **UniST** [69]: A universal STFM empowered by prompt learning, pretrained on multiple urban scenarios to achieve strong generalization. Official codebase (`https://github.com/tsinghua-fib-lab/UniST`) supports fixed horizon configurations only 6-step prediction. We retrain it on 13 datasets from the original release to support 12 and 96-step forecasting scenarios.

- **OpenCity** [33]: A versatile STFM that supports zero-shot and few-shot forecasting across diverse city-level applications. Integrated into our pipeline using the checkpoint `Opencity-plus.pth`, with adapter layers introduced to enable efficient few-shot adaptation to new datasets.

## A.4 More Results

### A.4.1 Model Size Analysis

As shown in Figure 9, we investigate the effect of model capacity by varying the number of Transformer layers in FactoST's temporal backbone (3.0M → 4.3M parameters) on ETTh2 long-term forecasting. Zero-shot performance improves steadily from 1 to 3 layers and then plateaus, indicating diminishing returns from deeper architectures. In the 10p few-shot setting, performance peaks at 3 layers and slightly degrades with 5–7 layers, with training logs revealing signs of overfitting—likely due to increased depth without proportional increases in regularization or hidden dimensionality. These results suggest that *moderate model capacity is optimal under data-limited conditions*, aligning with the principle of Occam's razor in transfer learning.

### A.4.2 Pretraining Data Scalability

Figure 8 shows the impact of pretraining corpus size on generalization. We train FactoST on 20% to 100% of Monash dataset and evaluate on ETTh2 long-term forecasting. Performance improves *monotonically* with more pretraining data, confirming that FactoST effectively leverages larger and more diverse temporal corpora. Notably, even at 100% data (13M points), FactoST remains far below the scale of leading foundation models (e.g., TimesFM: 100B), suggesting substantial headroom for improvement with access to richer and high quality pretraining sources.

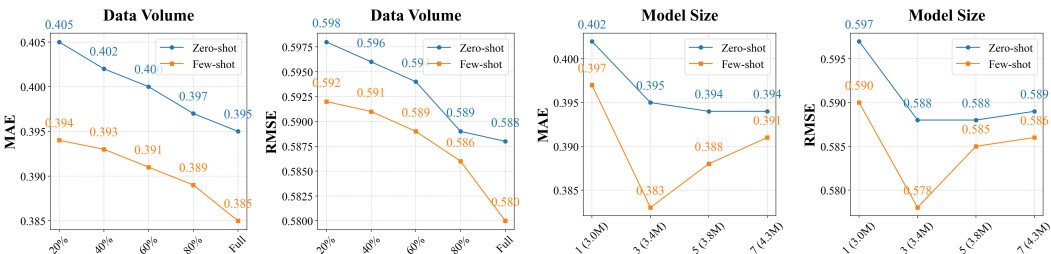

Figure 8: Pretraning data scaling analysis.   Figure 9: Model size analysis.

### A.4.3 Pretraining Objective Ablation

Our pretraining objective combines future prediction loss and spectral consistency loss with equal weighting: $\mathcal{L}_{\text{pretrain}} = \mathcal{L}_{\text{pred}} + \mathcal{L}_{\text{spec}}$. While $\mathcal{L}_{\text{pred}}$ captures temporal dynamics through supervised forecasting, $\mathcal{L}_{\text{spec}}$ enables self-supervised modeling of multi-frequency patterns via spectral consistency learning across frequency-isolated views. As shown in Table 8, removing $\mathcal{L}_{\text{spec}}$ degrades zero-shot MAE/RMSE by 3.54%/5.27%, but only slightly affects few-shot performance (↑0.78%/0.87%). This indicates that multi-frequency spectral consistency provides *complementary, non-redundant signals* that are especially crucial when no target-domain supervision is available. In zero-shot transfer, $\mathcal{L}_{\text{spec}}$ endows the model with robust spectral inductive biases that generalize across domains. During few-shot fine-tuning, limited labels allow the model to partially recover domain-specific patterns, reducing reliance on $\mathcal{L}_{\text{spec}}$. The modest few-shot gain likely reflects the moderate domain shift in our benchmarks; we expect $\mathcal{L}_{\text{spec}}$ to yield larger benefits under stronger distributional shifts (e.g., cross-city or cross-modality transfer).

Table 8: Ablation study on the pretraining objective ($\mathcal{L}_{\text{spec}}$) using ETTh2 long-term forecasting. Removing the spectral consistency loss degrades both zero-shot and few-shot performance.

| Model Variants | Zero-shot (MAE / RMSE) | Few-shot (MAE / RMSE) |
|---|---|---|
| FactoST | **0.395 / 0.588** | **0.383 / 0.578** |
| FactoST w/o $\mathcal{L}_{\text{spec}}$ | 0.409 / 0.619 | 0.386 / 0.583 |
| Δ (%) MAE / RMSE | +3.54% / +5.27% | +0.78% / +0.87% |

Table 9: Impact of Test-Time Computing (TTC) on long-term forecasting performance. FactoST with TTC consistently improves over the base model across all datasets.

| Dataset | FactoST | | FactoST w/ TTC | | Δ (%) | |
|---|---|---|---|---|---|---|
| | MAE | RMSE | MAE | RMSE | MAE | RMSE |
| PEMS-03 | 123.60 | 159.38 | **113.62** | **144.80** | -8.07% | -9.15% |
| PEMS-04 | 152.27 | 189.35 | **143.11** | **175.99** | -6.01% | -7.05% |
| PEMS-07 | 172.99 | 214.00 | **156.07** | **190.59** | -9.78% | -10.94% |
| PEMS-08 | 129.57 | 163.62 | **121.43** | **151.59** | -6.28% | -7.35% |
| PEMS-Bay | 7.10 | 12.01 | **6.12** | **10.47** | -13.80% | -12.82% |
| **Average** | 117.11 | 147.67 | **108.07** | **134.69** | -7.72% | -8.79% |

### A.4.4 Zero-Shot Enhancement

To address the inherent limitations of zero-shot forecasting, we integrate Test-Time Computing (TTC) [9] into FactoST—a lightweight online adaptation mechanism that refines predictions during inference without retraining. TTC maintains a FIFO memory queue of recent inputs, predictions, and (pseudo) labels, constructs a frequency-domain calibrator using FFT-based amplitude/phase offsets, and updates the calibrator using only historical predictions to avoid temporal leakage. As shown in Table 9, TTC consistently improves long-term zero-shot performance across datasets, reducing MAE and RMSE by 7.72% and 8.79% on average. Future work includes: (1) spatial meta-learning to capture universal topological priors (e.g., distance decay) [22]; (2) semantic-enhanced spatial embeddings via external knowledge (e.g., LLM-derived geolocation representations) [21]; and (3) cross-domain latent alignment to bridge spatial representation gaps [55].

### A.4.5 Temporal Feature Granularity Analysis

FactoST supports flexible multi-scale periodicity modeling through its spatio-temporal metadata fusion (STMF) module. While our main experiments use `time_of_day` (24) and `day_of_week` (7) identifiers—reflecting the dominant daily/weekly cycles in high-frequency traffic datasets like PEMS-03—the architecture readily accommodates longer-term patterns (e.g., monthly, yearly) by simply extending the temporal feature set, without any architectural changes. To validate this flexibility and assess the impact of temporal granularity, we replace daily/weekly features with `month_of_year` (12) in a few-shot setting on PEMS-03. As shown in Table 10, performance degrades significantly: short-term MAE increases by 12.94% and long-term MAE by 15.20%. This confirms that *temporal feature design must align with the intrinsic frequency of the data*—a principle our framework inherently supports through plug-and-play metadata injection.

Table 10: Impact of temporal feature granularity on PEMS-03 few-shot forecasting. Using coarse-grained monthly features harms performance due to mismatch with high-frequency data dynamics.

| Temporal Features | Short-term (MAE / RMSE) | Long-term (MAE / RMSE) |
|---|---|---|
| `time_of_day` & `day_of_week` | **17.54 / 28.10** | **28.57 / 46.78** |
| `month_of_year` | 19.81 / 31.37 | 32.91 / 52.01 |
| Δ (%) MAE / RMSE | +12.94% / +11.64% | +15.20% / +11.18% |

## B  Broader Impacts

Our work introduces a factorized framework for spatio-temporal foundation models that enhances efficiency, generalization, and cross-domain adaptability. By decoupling universal temporal pretraining from lightweight spatio-temporal adaptation, our approach significantly reduces computational cost and enables rapid few-shot deployment—making it well-suited for real-world applications with limited labeled data or constrained resources. The proposed method has the potential to benefit high-impact domains such as urban planning, traffic optimization, climate modeling, energy forecasting, and public health surveillance—areas where accurate, scalable, and transferable spatio-temporal prediction is crucial. Furthermore, the modular design promotes sustainable AI development by minimizing redundant large-scale pretraining and reducing overall energy consumption.

As this work primarily focuses on scientific research and technical innovation in spatio-temporal modeling, it does not present clear negative societal impacts. Instead, it contributes to the development of more accessible, efficient, and environmentally responsible foundation models for real world urban dynamics with both spatio and temporal characteristics.

## B.1 Limitations

Our work focuses on separating temporal pretraining from spatial adaptation to enable efficient and generalizable spatio-temporal modeling. While this factorized design offers strong empirical performance and flexibility, several limitations point to important directions for future research:

- **Spatial modeling impedes zero-shot generalization.** Our results reveal a key insight: explicit spatial modeling—especially when baked into pretraining—hurts cross-domain transfer because spatial structures (e.g., graph topology, sensor layout) are highly domain-specific. In contrast, temporal-only pretraining (as in FactoST 's UTP stage or TimesFM) achieves superior zero-shot performance, confirming that *universal temporal patterns*, not spatial priors, drive generalization. This explains why specialized STFMs (e.g., UniST, OpenCity) underperform or even fail catastrophically on unseen domains. While our factorized design avoids this pitfall by deferring spatial adaptation to the lightweight adapter, zero-shot spatio-temporal forecasting remains inherently challenging—particularly for long horizons—highlighting the need for complementary techniques (e.g., test-time computing [9]) to further bridge the domain gap.
- **Challenges with dynamic and open-world spatial structures.** The current framework assumes fixed node sets and static spatial topologies (e.g., traffic sensors or weather stations). It is not designed to handle scenarios where spatial units are added, removed, or reconfigured over time (e.g., mobile sensors, evolving infrastructure, or ad-hoc networks). Extending FactoST to support *zero-shot spatial generalization*—such as generating embeddings for unseen nodes or adapting to changing graph structures [31, 81]—remains an open challenge.
- **Dependency on pretraining corpus diversity.** Although our adapter-based design reduces computational overhead, the universal temporal backbone still relies on the *breadth and representativeness* of the pretraining data [75, 75]. In domains with strong physical laws or complex spatial couplings (e.g., power grids), the model may lack necessary inductive biases, limiting transferability.
- **Limited integration of exogenous variables.** Our current framework does not explicitly model external factors such as weather, events, or policy interventions—critical covariates in many real-world forecasting tasks [61, 67]. Developing mechanisms to incorporate and adapt to such *exogenous signals in a few-shot manner* is an important direction for enhancing practical utility.
- **Static adapter composition.** The adapter $S$ currently applies a fixed set of modules (STMF, STF, HDA, CMR) regardless of input characteristics. A more intelligent system could enable *adaptive model composition*: by analyzing temporal stability, spatial heterogeneity, or domain shift, it could dynamically choose between full spatio-temporal modeling, temporal-only inference, or specialized lightweight modules—improving both robustness and efficiency [50].
- **Suboptimal fine-tuning protocols.** Current adaptation uses uniform gradient updates on the adapter, which may underutilize pretrained knowledge and risk catastrophic forgetting. Parameter-efficient strategies—such as prompt tuning, selective layer retraining, or regularization-aware updates—could better preserve temporal priors while enabling efficient spatial adaptation [45, 7].

These limitations point to several promising directions for future work: (1) rethinking spatial modeling to avoid domain-specific biases in pretraining, (2) enabling adaptation to dynamic or unseen spatial configurations, (3) enriching temporal foundations with exogenous context, (4) leveraging more efficient and stable fine-tuning strategies, and (5) developing hybrid inference mechanisms (e.g., test-time adaptation) to bridge the zero-shot performance gap. Addressing these challenges will be key to building truly robust, scalable, and practical spatio-temporal foundation models.

## B.2 Pseudocode of FactoST

For reproducibility, we present the detailed pseudocode of FactoST in Algorithm 1, which concisely summarizes the two-stage learning paradigm: universal temporal pretraining (UTP) followed by lightweight spatio-temporal adaptation (STA).

**Algorithm 1** FACTOST: Factorized Spatio-Temporal Foundation Model

---

**Require:** Cross-domain ST datasets $\mathcal{D} = \{(\mathbf{X}^{(j)}, \mathbf{Y}^{(j)}, \mathbf{m}^{(j)})\}_j$, where $\mathbf{X}^{(j)} \in \mathbb{R}^{N_j \times L \times D}$, $\mathbf{Y}^{(j)} \in \mathbb{R}^{N_j \times F \times D}$, and $\mathbf{m}^{(j)}$ denotes ST metadata.

1: // **Stage I: Universal Temporal Pretraining (UTP)**
2: Initialize temporal backbone $T$ and domain prompts $\mathbf{p} \in \mathbb{R}^{K_p \times d}$
3: **for** each node-wise sequence $\mathbf{x} \in \mathbb{R}^{L \times D}$ sampled from $\mathcal{D}$ **do**
4:     // Multi-frequency augmentation
5:     $\mathbf{x}_f \leftarrow \text{FFT}(\mathbf{x})$
6:     **for** $i = 1$ to $K_m$ **do**
7:         Sample $\tau_i \sim \text{Uniform}(0, \lfloor L/2 \rfloor + 1)$, $\mu_i \sim \text{Bernoulli}(p)$
8:         Mask $\mathbf{x}_f^{(i)} \leftarrow \begin{cases} \mathbf{x}_f[\tau_i :] = 0 & \text{if } \mu_i = 0 \\ \mathbf{x}_f[: \tau_i] = 0 & \text{if } \mu_i = 1 \end{cases}$
9:         $\mathbf{x}_{\text{m}}^{(i)} \leftarrow \text{iFFT}(\mathbf{x}_f^{(i)})$
10:     **end for**
11:     $\mathbf{x}_{\text{aug}} \leftarrow [\mathbf{x}, \{\mathbf{x}_{\text{m}}^{(i)}\}_{i=1}^{K_m}]$; apply patching $\rightarrow \mathbb{R}^{K_m \times N' \times L' \times d}$
12:     // Multi-domain prompting
13:     $\mathbf{x}_e \leftarrow \text{Linear}(\text{Patch}(\mathbf{x})) \in \mathbb{R}^{N_p \times d}$
14:     Compute attention: $s_j = -\|\mathbf{x}_e - \mathbf{p}_j\|_2^2$, $\alpha_j = \text{softmax}(s_j)$
15:     $\mathbf{x}_p \leftarrow \sum_{j=1}^{K_p} \alpha_j \mathbf{p}_j$; form $\mathbf{x}_{\text{ctx}} = [\text{Patch}(\mathbf{x}), \mathbf{x}_p]$
16:     // Multi-task pretraining
17:     $L_{\text{recon}} \leftarrow \|\mathbf{x} - \text{ReconHead}(T_{\text{dec}}(T_{\text{enc}}(\mathbf{x}_{\text{aug}})))\|_2^2$
18:     $\hat{\mathbf{y}} \leftarrow \text{PredHead}(T_{\text{dec}}(T_{\text{enc}}(\mathbf{x}_{\text{ctx}})))$
19:     $L_{\text{pred}} \leftarrow \|\mathbf{y} - \hat{\mathbf{y}}\|_2^2 + \|\mathbf{x}_e - \mathbf{x}_p\|_2^2$
20:     Update $T$ and $\mathbf{p}$ with $L = L_{\text{recon}} + L_{\text{pred}}$
21: **end for**
22: // **Stage II: Spatio-Temporal Adaptation (STA)**
23: Initialize lightweight adapter $S$ with modules: STMF, STF, HDA, CMR
24: Freeze $T$; initialize memory buffer $\mathcal{M} \leftarrow \emptyset$
25: **for** each mini-batch $\{(\mathbf{X}, \mathbf{Y}, \mathbf{m})\} \subset \mathcal{D}$ **do**
26:     // Temporal feature extraction
27:     $\mathbf{z} \leftarrow T(\mathbf{X}) \in \mathbb{R}^{N \times N' \times d}$ {via patching and encoder}
28:     // STMF: inject ST metadata
29:     $\mathbf{H}_{\text{st}} \leftarrow \text{Proj}\big([\mathbf{E}_n^i \,\|\, \{\mathbf{E}_c^{\phi_c(\tau)}\}_{c \in \mathcal{S}}]\big) \in \mathbb{R}^{N \times N' \times d}$
30:     $\mathbf{H}_{\text{fused}} \leftarrow \mathbf{z} + \mathbf{H}_{\text{st}}$
31:     // STF: adaptive filtering
32:     Compute low-rank affinities: $\mathbf{S}_s = \langle \mathbf{H}_{\text{st}} \mathbf{W}_q^{(s)}, \mathbf{E}_n \mathbf{W}_k^{(s)} \rangle$, $\mathbf{S}_t = \langle \mathbf{H}_{\text{st}} \mathbf{W}_q^{(t)}, \mathbf{E}_t \mathbf{W}_k^{(t)} \rangle$
33:     $\mathbf{S}_d \leftarrow \sum_{\delta=1}^{\Delta} \gamma^{(\delta)} \cdot \langle \mathbf{H}_{\text{fused}}, \text{Agg}_\delta(\mathbf{H}_{\text{fused}}^{(t-\delta)}) \rangle$
34:     $\mathbf{W} \leftarrow \text{softmax}\big([\mathbf{S}_s, \mathbf{S}_t, \mathbf{S}_d] \mathbf{W}_{\text{att}} / \tau_{\text{att}}\big)$
35:     $\mathbf{H}_{\text{refined}} \leftarrow \text{LayerNorm}(\mathbf{H}_{\text{fused}} \odot \mathbf{W})$
36:     // HDA: hierarchical alignment
37:     $\mathcal{K} \leftarrow \text{Topk}_j(-\|\mathbf{x}_e - \mathbf{p}_j\|_2)$, $\bar{\mathbf{p}}_k \leftarrow \frac{1}{k} \sum_{j \in \mathcal{K}} \mathbf{p}_j$
38:     $\mathbf{A} \leftarrow \mathbf{u}\mathbf{v}^\top$, $\mathbf{X}_r \leftarrow (\mathbf{1}_{N_p} \bar{\mathbf{p}}_k^\top) \odot \mathbf{A}$
39:     $\mathbf{H}_{\text{aligned}} \leftarrow \mathbf{H}_{\text{refined}} + \text{Proj}(\mathbf{X}_r)$
40:     // CMR: continual replay
41:     Update memory buffer $\mathcal{M}$ (e.g., FIFO or reservoir sampling)
42:     Sample replay batch $\mathcal{B}_r \subset \mathcal{M}$, mix with current batch $\mathcal{B}_c$
43:     $\mathcal{B} \leftarrow \mathcal{B}_c \cup \mathcal{B}_r$
44:     // Adapter update
45:     $\mathbf{Y}_{\text{out}} \leftarrow S(\mathbf{H}_{\text{aligned}}; \mathbf{m})$
46:     Update $S$ with $\|\mathbf{Y} - \mathbf{Y}_{\text{out}}\|_2^2$ on $\mathcal{B}$
47: **end for**
48: **return** Final model: $\mathcal{F}(\mathbf{X}; \mathbf{m}) = S(T(\mathbf{X}); \mathbf{m})$

