# OpenReview forum: "Learning to Factorize Spatio-Temporal Foundation Models"
_NeurIPS.cc/2025/Conference — NeurIPS 2025 spotlight_

### Official Review · Reviewer_UUgZ · 2025-06-18

**Clarity:** 3
**Significance:** 2
**Originality:** 3
**Rating:** 5
**Confidence:** 4

**Summary:**

The paper proposes a two-stage factorized paradigm that decouples Universal Temporal Pretraining(UTP) from Spatio-Temporal Adaptation(STA). The first stage UTP learns general, domain-invariant temporal patterns through a graph-agnostic and scalable temporal backbone, enhanced by multi-frequency augmentation and domain-aware prompts. In the second stage, a compact adapter is introduced—either with the UTP backbone frozen or fine-tuned—to inject spatial awareness and domain specificity through metadata-aware token encoding, sparse low-rank dependency modeling, hierarchical prompt alignment, and replay-based training stabilization. This decoupling enables efficient learning and adaptation while preserving strong temporal capabilities without the need for costly joint spatio-temporal pretraining.

**Questions:**

1. Could the authors explicitly clarify that the specific research focus of this work is spatio-temporal forecasting? Moreover, the use of the term "factorized" is a bit unclear. It’s not obvious how the proposed two-stage framework reflects the idea of factorization. The pre-training and fine-tuning setup is a commonly used paradigm, and without a more formal or concrete explanation of how factorization is involved, the term might feel overstated.

2. Is the second-stage SAT method exclusively applicable to GNN-based backbones? Has it been adapted to other types of architectures, such as the more commonly used Encoder-Decoder structures[1] in spatio-temporal tasks? If the method is only applicable to GNNs, I am concerned that the generalizability of this work may be limited.

3. Please include more expert models [2] for a more comprehensive evaluation. Please explain why the proposed method does not achieve state-of-the-art performance on certain datasets. Is the dataset configuration used in the paper aligned with widely accepted evaluation protocols? Could the authors discuss the impact of the number of pre-training datasets on downstream performance, and provide the criteria used for dividing the pre-training and fine-tuning datasets?

4. Could the authors explain why only daily and weekly periodicity are considered, while quarterly and yearly periodicities are not taken into account?

5. Generally, smaller models tend to have faster inference speeds; so why does FactoST, despite having only a fraction of the parameters of Informer and FEDformer, exhibit slower inference speed than these models?


References:

[1] Tan, C., Li, S., Gao, Z., Guan, W., Wang, Z., Liu, Z., Wu, L. and Li, S.Z., 2023. Openstl: A comprehensive benchmark of spatio-temporal predictive learning. Advances in Neural Information Processing Systems, 36, pp.69819-69831.

[2] Wang, Y., Wu, H., Dong, J., Liu, Y., Long, M. and Wang, J., 2024. Deep time series models: A comprehensive survey and benchmark. arXiv preprint arXiv:2407.13278.

**Ethical Concerns:**

["NO or VERY MINOR ethics concerns only"]

**Final Justification:**

The paper proposes a two-stace factorized paradiom that decouples universal temporal pretraining from spatio-temporal adaptations. This decouping enables eficient learning and adaptation while presering strong temporal capabilities without the need for costly joint spatio-temporal pretraining. The authors' response has adequately addressed my concerns, and I support the acceptance of the paper.

**Limitations:**

The authors mention the limitations of their approach in the zero-shot setting; however, they do not address the fact that their validation is restricted solely to spatio-temporal forecasting tasks, nor that fine-tuning is conducted exclusively on the GNN backbone.

**Paper Formatting Concerns:**

There are no formatting issues.

**Quality:**

3

**Strengths And Weaknesses:**

Strengths:
1. The paper proposes a novel foundational pre-training and fine-tuning paradigm for time series forecasting. It conducts pre-training on datasets from diverse domains and validates its effectiveness across a wide range of time series forecasting tasks.

2. The proposed model, FactoST, achieves significantly improved performance compared to existing Spatio-Temporal Foundation Models (STFMs). For instance, FactoST outperforms previous STFM approaches by 46.4% in Mean Absolute Error (MAE), highlighting its potential in enhancing model accuracy and robustness.

3. I believe this work is somewhat meaningful and would benefit from further clarification and discussion of its scope, generalizability, and technical novelty, particularly in its attempt to unify pre-training strategies across heterogeneous time series datasets.


Weaknesses:
1. I feel that the scope of the paper is somewhat too broad, and the authors should better define a more specific research focus. Since the paper mentions spatio-temporal modeling, it would be reasonable to expect coverage of various tasks, such as spatio-temporal classification, not just forecasting. However, the current experiments are limited to forecasting tasks. It would be helpful if the authors could clearly state that the focus of this work is on spatio-temporal forecasting. Moreover, the use of the term "factorized" is a bit unclear. It’s not obvious how the proposed two-stage framework reflects the idea of factorization. The pre-training and fine-tuning setup is a commonly used paradigm, and without a more formal or concrete explanation of how factorization is involved, the term might feel overstated.

2. Is the second-stage SAT method exclusively applicable to GNN-based backbones? Has it been adapted to other types of architectures, such as the more commonly used Encoder-Decoder structures[1] in spatio-temporal tasks? If the method is only applicable to GNNs, I am concerned that the generalizability of this work may be limited.

3. There are several issues in the experimental section. First, the baseline comparisons are insufficient; it is recommended to include more expert models [2] for a more comprehensive evaluation. Second, for datasets where the proposed method does not achieve state-of-the-art performance, no reasonable analysis is provided. Regarding the experimental setup, the criteria for dividing the pre-training and fine-tuning datasets are unclear. Furthermore, the impact of the number of pre-training datasets on performance is not discussed. Is the dataset configuration used in the paper aligned with widely accepted evaluation protocols? If so, the authors should clarify this and provide justification.

4. In Section 3.2, Stage II: Spatio-Temporal Adaptation, the Spatio-Temporal Metadata Fusion (STMF) approach, where periodicity is modeled only for daily and weekly cycles, is unreasonable. Daily and weekly periodicity typically captures short-term fluctuations and repeating patterns. However, in real-world applications, many data are also influenced by long-term trends (such as inter-annual variations) and multiple periodicities (e.g., quarterly, yearly). For example, certain industries (such as tourism, electricity consumption, etc.) may exhibit seasonal changes or be impacted by economic activities at different times of the year. Can these factors truly be captured solely by modeling daily and weekly periodicity?

5. Generally, the smaller the model's parameter size, the faster the inference speed. However, in Figure 5, FactoST has only a fraction of the parameters of Informer and FEDformer, yet its inference speed is still slower than theirs. The paper does not provide a reasonable explanation for this phenomenon.


References:

[1] Tan, C., Li, S., Gao, Z., Guan, W., Wang, Z., Liu, Z., Wu, L. and Li, S.Z., 2023. Openstl: A comprehensive benchmark of spatio-temporal predictive learning. Advances in Neural Information Processing Systems, 36, pp.69819-69831.

[2] Wang, Y., Wu, H., Dong, J., Liu, Y., Long, M. and Wang, J., 2024. Deep time series models: A comprehensive survey and benchmark. arXiv preprint arXiv:2407.13278.

---

> ### Author Rebuttal · Authors · 2025-07-30
>
> Thank you for your insightful questions and suggestions. We have revised and expanded our explanations to address your concerns.
>
> > **Q1 (Weakness - Scope/Clarity): Could the authors explicitly clarify that the specific research focus of this work is forecasting? Moreover, the term "factorized" is unclear—how does the two-stage framework reflect factorization?**
> >
>
> **A1:** Yes, FactoST is specifically designed for spatio-temporal forecasting. The term **factorized** refers to our core innovation: unlike existing STFMs that perform **joint** spatio-temporal **pretraining**, FactoST **decouples** learning into two sequential stages—first extracting general temporal patterns (UTP), then injecting domain-specific spatial context (STA). This separation explicitly factorizes temporal and spatial learning, in contrast to standard pretraining/fine-tuning where both dimensions are entangled from the start.
>
> As shown in Table 1, this factorization offers three key advantages: (1) UTP eliminates spatial parameters, **reducing** pretraining **cost**; (2) STA adapters constitute only **a small fraction** of total parameters, enabling **fast** adaptation; (3) the framework maintains cross-domain **versatility** while achieving superior few-shot performance. Figure 1(c) illustrates how this design **avoids** the spatial generalization bottleneck of joint learning approaches.
>
> > **Q2 (Weakness - Generalizability): Is the STA method limited to GNN-based backbones? Has it been tested on other types of architectures, such as the more common encoder-decoder structures? If restricted to GNNs, the generalizability may be limited.**
> >
>
> **A2:** No, **STA is architecture-agnostic** and does **not require** GNN backbones. It operates at the feature level, using ST metadata fusion, dynamic ST filtering, hierarchical domain alignment, and continual memory replay to offer a **lightweight** spatio-temporal modeling solution. This design allows it to work with any temporal backbone architecture, as it processes feature embeddings **regardless** of the source model.
>
> To validate this, we integrated STA into **PatchTST**, a non-GNN, time-series-focused model, and evaluated its few-shot performance on long-term tasks. The results (below) show that **PatchTST w/ STA** outperforms vanilla PatchTST across all datasets, confirming STA’s ability to generalize beyond GNNs. FactoST itself still achieves superior results, which we attribute to its pretrained temporal backbone’s universal pattern-learning capabilities. This demonstrates STA’s broad compatibility and strengthens the generalizability of our framework.
>
> | **Dataset** | **FactoST (MAE/RMSE)** | **PatchTST w/ STA (MAE/RMSE)** | **PatchTST (MAE/RMSE)** |
> | --- | --- | --- | --- |
> | PEMS-03 | 28.57 / 46.78 | 31.48 / 49.67 | 61.22 / 100.33 |
> | PEMS-04 | 42.04 / 64.89 | 43.69 / 65.17 | 70.71 / 104.00 |
> | PEMS-07 | 45.60 / 72.47 | 51.75 / 80.12 | 80.09 / 118.54 |
> | PEMS-08 | 35.69 / 56.15 | 44.00 / 66.85 | 57.31 / 87.33 |
> | PEMS-Bay | 2.96 / 6.21 | 3.57 / 7.72 | 4.32 / 9.22 |
> | ETTh2 | 0.358 / 0.561 | 0.362 / 0.569 | 0.943 / 1.609 |
> | Electricity | 0.265 / 0.409 | 0.279 / 0.414 | 0.442 / 2.747 |
> | Weather | 0.226 / 0.426 | 0.223 / 0.419 | 0.708 / 1.409 |
>
> > **Q3 (Weakness - Experiments/Baselines): Please include more expert models for a comprehensive evaluation. Why doesn’t the method achieve SOTA on all datasets? Are the dataset protocols standard? How does pretraining data volume affect downstream performance?**
> >
>
> **A3:**
>
> - **Additional Baselines:** As suggested, we include two recent baselines (TimeMixer, BigST) for a more comprehensive comparison. Results (below) show FactoST **tops** seven of the eight datasets, demonstrating its **competitiveness**. This aligns with our core finding that the factorized UTP+STA design outperforms both specialized and foundation models in most scenarios.
>
>
>     | **Dataset** | **FactoST (MAE/RMSE)** | **BigST (MAE/RMSE)** | **TimeMixer (MAE/RMSE)** |
>     | --- | --- | --- | --- |
>     | PEMS-03 | 28.57 / 46.78 | 51.87 / 75.56 | 47.86 / 71.52 |
>     | PEMS-04 | 42.04 / 64.89 | 52.37 / 80.23 | 58.44 / 86.67 |
>     | PEMS-07 | 45.60 / 72.47 | 54.92 / 82.12 | 67.75 / 100.45 |
>     | PEMS-08 | 35.69 / 56.15 | 58.68 / 86.76 | 45.10 / 65.53 |
>     | PEMS-Bay | 2.96 / 6.21 | 2.93 / 6.20 | 4.11 / 8.94 |
>     | ETTh2 | 0.358 / 0.561 | 1.164 / 1.797 | 1.013 / 1.646 |
>     | Electricity | 0.265 / 0.409 | 0.481 / 2.843 | 0.459 / 2.833 |
>     | Weather | 0.087 / 0.276 | 0.892 / 1.522 | 0.720 / 1.401 |
>
>     We will cite the references you provided ([1] OpenSTL by Tan et al., 2023; [2] Deep Time Series Models by Wang et al., 2024) in the revised manuscript and incorporate their insights into our discussion of related work and future directions.
>
> - **Performance Gap Analysis:** The minor performance gaps on specific datasets highlight important trade-offs in model design. On the **PEMS-07** (short-term), FactoST slightly underperforms D2STGNN, which explicitly decouples traffic flow into diffusion and inherent signals—a fine-grained modeling approach highly effective for its dense topology. However, this comes at high computational cost, causing OOM failures in long-term forecasting and limiting practicality. On datasets like **ETTh2** that lack spatial structure, Moirai performs better as it is is specifically optimized for general time series without spatial priors, while FactoST is designed for spatio-temporal settings. Nevertheless, FactoST uses significantly fewer parameters (1.4% of Moirai’s), runs faster, and still outperforms other STFMs such as UniST and OpenCity. Future work will explore lightweight spatial modeling for complex scenarios, with further discussions in Q2/A2 and Q4/A4 of Reviewer jNeH.
> - **Dataset Protocol:** We follow strict domain separation. UTP pretraining uses the Monash dataset collection; evaluation uses standard benchmarks (PEMS, etc.) with the widely adopted 6:2:2 train/validation/test split. For few-shot settings, we sample 10% of training data, resulting in a 0.6:2:2 effective split.
> - **Pretraining Data Impact:** To evaluate scalability, we conducted an ablation study on ETTh2 (long-term forecasting), varying the UTP pretraining data volume from 20% to 100%. Results show that performance improves **monotonically** with more data, confirming FactoST benefits from larger pretraining corpora. Notably, our current dataset (13M points) is much smaller than those used by existing foundation models (see Q4/A4 of Reviewer rPHb), indicating potential for further improvement with more high-quality data.
>
>
>     | **Data Volume** | **Zero-shot (MAE/RMSE)** | **Few-shot (MAE/RMSE)** | **Δ (%) Zero-shot vs. Full** | **Δ (%) Few-shot vs. Full** |
>     | --- | --- | --- | --- | --- |
>     | 20% | 0.405 / 0.598 | 0.394 / 0.592 | +2.53% / +1.70% | +2.34% / +1.72% |
>     | 40% | 0.402 / 0.596 | 0.393 / 0.591 | +1.77% / +1.36% | +1.82% / +1.55% |
>     | 60% | 0.400 / 0.594 | 0.391 / 0.589 | +1.27% / +1.02% | +1.30% / +1.21% |
>     | 80% | 0.397 / 0.589 | 0.389 / 0.586 | +0.51% / +0.17% | +0.78% / +0.69% |
>     | Full | 0.395 / 0.588 | 0.385 / 0.580 | - / - | - / - |
>
> > **Q4 (Weakness - Model Design): Could the authors explain why only daily and weekly periodicity are considered, while quarterly and yearly periodicities are not taken into account?**
> >
>
> **A4:** FactoST supports **flexible multi-scale** periodicity modeling. The current implementation focuses on daily/weekly cycles because they are dominant in the high-frequency traffic and energy datasets used for evaluation (e.g., 5-minute traffic data in PEMS-03). However, the architecture is designed to be general and can incorporate longer-term patterns (like monthly or annual cycles) by utilizing timestamp features. The STMF module can encode longer cycles by expanding time-of-day/day-of-week embeddings to include month-of-year/yearly features, with no architectural changes. We will **clarify this generality** in the revised manuscript.
>
> To validate this flexibility, we conducted a few-shot experiment on PEMS-03 using "month-of-year" features instead of daily/weekly cycles. As shown below, performance **degrades significantly** (short-term MAE ↑12.94%, long-term MAE ↑15.20%), due to a **mismatch** between the dataset’s high temporal frequency and the long-term nature of monthly features. This outcome confirms that temporal feature selection must align with the data's inherent frequency—a design principle our framework **inherently supports**.
>
> | **Temporal features** | **short-term (MAE/RMSE)** | **long-term (MAE/RMSE)** |
> | --- | --- | --- |
> | time_of_day:24,day_of_week:7 | 17.54 / 28.10 | 28.57 / 46.78 |
> | month_of_year:12 | 19.81 / 31.37 | 32.91 / 52.01 |
> | Δ (%) MAE/RMSE | +12.94% / +11.64% | +15.20% / +11.18% |
>
> > **Q5 (Weakness - Efficiency): Smaller models usually have faster inference—why is FactoST slower than larger models like Informer and FEDformer?**
> >
>
> **A5:** While FactoST has fewer parameters than models like Informer/FEDformer, its inference is only slightly slower due to the necessary computations in the STA module—spatio-temporal metadata fusion, sparse interaction modeling via filtering, and hierarchical domain alignment—which adapt the universal temporal backbone to specific spatial contexts. This modest cost is justified by the substantial performance **gains** and the fact that FactoST remains significantly **faster** than joint STFMs (OpenCity, UniST). The trade-off strongly favors FactoST, as it achieves the **best** overall performance-efficiency **balance** among the evaluated models (Figure 5).
>
> ---
>
> We are deeply grateful for your comprehensive feedback, which has greatly helped us strengthen the paper's clarity and rigor. We will carefully revise the manuscript to include the additional experiments and clarifications you suggested.

---

> ### Comment · Reviewer_UUgZ · 2025-08-06
>
> I acknowledge the authors' response to my comments. The rebuttal has adequately addressed my concerns, and I support the acceptance of the paper.

---

### Official Review · Reviewer_jNeH · 2025-06-30

**Clarity:** 4
**Significance:** 4
**Originality:** 4
**Rating:** 5
**Confidence:** 5

**Summary:**

FactoST proposes a two-stage factorized learning framework for Spatio-Temporal Foundation Models (STFMs), decoupling universal temporal pretraining (UTP) from domain-specific spatio-temporal adaptation (STA). The first stage learns a spatially-agnostic temporal backbone via multi-frequency reconstruction and domain-aware prompting. The second stage introduces a lightweight adapter that injects spatial awareness through metadata fusion, sparse interaction modeling, hierarchical domain alignment, and continual memory replay. Experimental results show that FactoST outperforms existing STFMs in both accuracy and efficiency across multiple domains.

**Questions:**

1. The UTP dataset spans six major domains, but data quantity and quality may vary significantly across them (e.g., energy vs. health). Have the authors investigated how such imbalances affect generalization performance?
2. The paper notes the limited zero-shot capabilities of current STFMs. While FactoST focuses on few-shot adaptation, can the authors suggest directions for improving zero-shot transfer?

**Ethical Concerns:**

["NO or VERY MINOR ethics concerns only"]

**Final Justification:**

The author's rebuttal addressed all of my concerns. I will maintain my current positive score and support its acceptance.

**Limitations:**

The paper briefly mentions limitations related to dependency on the pre-training corpus and handling evolving spatial structures. A more thorough discussion on how the framework might be extended to address these challenges would strengthen the work and guide future research.

**Paper Formatting Concerns:**

N\A

**Quality:**

4

**Strengths And Weaknesses:**

**Strengths:**

- As the first work to explicitly decouple temporal and spatial learning in STFMs, this approach introduces a conceptually clear and potentially influential paradigm for future research.
- The method strikes a good balance between building a general-purpose temporal backbone and enabling efficient, domain-specific adaptation — making it well-suited for complex real-world spatio-temporal scenes.
- Experimental results are consistently strong across diverse domains and forecasting tasks. Ablation studies provide clear insight into the role of key components.
- The paper is clearly written, with effective visualizations that aid in conveying the model architecture and workflow.

**Weaknesses:**

- The impact of domain imbalance or data quality variations in the UTP phase is not analyzed, which could affect the robustness of the learned representations.
- While the proposed adapter improves efficiency, the potential gains from more complex spatial modeling are not fully explored.

---

> ### Author Rebuttal · Authors · 2025-07-30
>
> Thank you for recognizing our work. Below, we hope to resolve some of your doubts.
>
> > **Q1: The UTP dataset spans six major domains, but data quantity and quality may vary significantly across them (e.g., energy vs. health). Have the authors investigated how such imbalances affect generalization performance?**
> >
>
> **A1:** FactoST addresses potential domain imbalance through **dual** strategies:
>
> - **Data-level:** For domains with limited data, multi-frequency augmentation is applied to synthetically **expand temporal patterns**. Consistent preprocessing (outlier removal, gap handling) is applied across all data to ensure quality consistency.
> - **Model-level:** A multi-domain prompting mechanism (Equation 2) dynamically adjusts the influence of different domains using **adaptive** weights (α_j). Ablation experiments (Figure 3) and visualizations (Figure 6) demonstrate its **effectiveness** in learning domain-specific knowledge, **mitigating** the impact of imbalances and improving generalization.
>
> > **Q2: The paper notes the limited zero-shot capabilities of current STFMs. While FactoST focuses on few-shot adaptation, can the authors suggest directions for improving zero-shot transfer?**
> >
>
> **A2:** To explore directions for improving zero-shot performance, we integrated the **Test-Time Computing (TTC)** method proposed in [1] into our framework. This approach performs **online adaptive optimization** during the **testing** phase to correct model predictions. It works as follows:
>
> 1. It maintains a FIFO memory queue to cache gradually predicted inputs, predictions, and ground-truth labels during testing;
> 2. A calibrator is constructed to transform queue-based historical predictions into the frequency domain via FFT, using group-learned amplitude/phase offsets to dynamically correct non-stationary temporal deviations;
> 3. Loss is calculated solely from queue-sampled historical predictions to update calibrator parameters, strictly preserving temporal consistency to prevent information leakage.
>
> The integration of TTC **effectively** boosts long-term zero-shot performance, as shown in the results below. On average, it **reduces** MAE and RMSE by 7.72% and 8.79%, respectively, demonstrating that online adaptation is a viable direction.
>
> | Dataset | FactoST (MAE/RMSE) | FactoST w/ TTA (MAE/RMSE) | Δ (%) MAE / RMSE |
> | --- | --- | --- | --- |
> | PEMS-03 | 123.60 / 159.38 | 113.62 / 144.80 | -8.07% / -9.15% |
> | PEMS-04 | 152.27 / 189.35 | 143.11 / 175.99 | -6.01% / -7.05% |
> | PEMS-07 | 172.99 / 214.00 | 156.07 / 190.59 | -9.78% / -10.94% |
> | PEMS-08 | 129.57 / 163.62 | 121.43 / 151.59 | -6.28% / -7.35% |
> | PEMS-Bay | 7.10 / 12.01 | 6.12 / 10.47 | -13.80% / -12.82% |
> | Average | 117.11 / 147.67 | 108.07 / 134.69 | -7.72% / -8.79% |
>
> Future directions for zero-shot improvement include: (1) spatial meta-learning to encode universal topological features like distance decay [2]; (2) semantic-enhanced spatial priors using external knowledge [3]; (3) cross-domain latent space alignment to bridge spatial representation gaps [4].
>
> [1] Learning with Calibration: Exploring Test-Time Computing of Spatio-Temporal Forecasting, arxiv 2025
>
> [2] SPOK: tokenizing geographic space for enhanced spatial reasoning in GeoAI, IJGIS 2025
>
> [3] Geolocation Representation from Large Language Models are Generic Enhancers for Spatio-Temporal Learning, AAAI 2025
>
> [4] Unveiling the Inflexibility of Adaptive Embedding in Traffic Forecasting, Arxiv 2024
>
> > **Q3 (Weakness): While the proposed adapter improves efficiency, the potential gains from more complex spatial modeling are not fully explored.**
> >
>
> **A3:** STA employs **feature-level** dependency modeling: it dynamically identifies key correlations and suppresses noise through interactions between input features and learnable identifiers, while reducing computational complexity via low-rank projection and sparse strategies.
>
> Its advantages include: simultaneously capturing spatial proximity (S_s), temporal coherence (S_t), and delayed causalities (S_d), breaking the limitation of traditional models that only model single or two dimensions; dynamically adjusting the contribution weights of spatial, temporal, and delayed relationships through learnable parameters to adapt to different data characteristics.
>
> Complex spatial modeling remains underexplored, as more sophisticated graph modeling mechanisms (e.g., D2STGNN) incur **extremely high** resource consumption (OOM) in long-term prediction (Table 3), **restricting practical usability**. As shown in our added experiment (**Q2/A2 of Reviewer UUgZ Q2**), the current STA design is already **effective** and **architecture-agnostic**, offering a good **balance**. Future work will explore more lightweight methods for modeling high-order spatial dependencies.
>
> > **Q4 (Limitation Discussion): The paper briefly mentions limitations related to dependency on the pretraining corpus and handling evolving spatial structures. A more thorough discussion would strengthen the work.**
> >
>
> **A4:** We appreciate this valuable feedback. Addressing the limitations related to pretraining corpus dependency and evolving spatial structures is crucial for future work. Our current framework offers several avenues for extension:
>
> 1. **Integration of Exogenous Variables**: Enhancing FactoST to effectively integrate and adapt to external factors (e.g., weather, events) in a few-shot manner is an important direction, as this capability remains underexplored in STFMs.
> 2. **Handling Dynamic Spatial Nodes:** Future research could focus on making FactoST robust to real-time changes in the number and identity of spatial nodes (e.g., sensors added/removed), ensuring generalization to new environments while preserving pretrained knowledge.
> 3. **Adaptive Model Composition**: Leveraging the factorized design, FactoST could be extended to adaptively select components based on data characteristics. By evaluating temporal stability and spatial distinctiveness, the system could dynamically choose between full spatio-temporal modeling, temporal-only modeling, or specialized modules for highly dynamic scenarios, enhancing robustness.
>
> ---
>
> We sincerely appreciate your thoughtful questions and valuable suggestions on future directions. We will expand the discussion on zero-shot learning and model limitations as recommended.

---

> > ### Comment · Reviewer_jNeH · 2025-08-05
> >
> > The author's rebuttal addressed all of my concerns. I will maintain my current positive score and support its acceptance.

---

### Official Review · Reviewer_cLs2 · 2025-07-01

**Clarity:** 3
**Significance:** 3
**Originality:** 3
**Rating:** 5
**Confidence:** 3

**Summary:**

The paper focuses on developing a spatio-temporal foundation model (STFM) that builds a unified model to capture spatial correlations across domains (e.g., power grids and urban road networks) while jointly learning spatial and temporal dependencies in a computationally efficient manner. In particular, the paper proposes a two stage paradigm for STFM, FactoST that decouples universal temporal pretraining (UTP) from spatio-temporal adaptation (STA).

UTP aims to learn universal temporal patterns across diverse domains through multi-frequency augmentation (encouraging the model to align multi-frequency information across scales), multi-domain prompting (guiding the model with task-specific context using explicit spatial graphs), and multi-task pretraining (combining self-supervised reconstruction and supervised forecasting objectives). ​​STA finetunes the UTP backbone and adds four lightweight, compact adapters to effectively capture spatio-temporal dependencies. ST metadata fusion is used for spatial-aware feature alignment, ST filtering for sparse interaction modeling, hierarchical domain alignment to bridge domain gaps, and continual memory replay to mitigate knowledge forgetting.

Extensive evaluations in a few-shot setting, including comparative and ablation studies, demonstrate the effectiveness of the proposed method compared to domain-specific expert models.

**Questions:**

Please refer to the points listed in the Weaknesses.

**Ethical Concerns:**

["NO or VERY MINOR ethics concerns only"]

**Final Justification:**

Most of my concerns regarding the contribution of ```L_pred``` and ```L_recon``` during pre-training, clarification on zero-shot evaluation and stability claims have been satisfactorily resolved. Therefore, I will raise my current score and support its acceptance.

**Limitations:**

Yes

**Quality:**

3

**Strengths And Weaknesses:**

Strengths

1. The paper identifies two key challenges in developing a spatio-temporal foundation model (STFM). First, spatial correlations vary across domains and second, jointly learning spatial and temporal dependencies is computationally expensive. To address these issues, the paper proposes a novel method, FactoST, which decouples universal temporal pretraining from spatio-temporal adaptation. This enables efficient learning and adaptation while preserving strong temporal capabilities without requiring full spatio-temporal pretraining.

2. The paper is technically sound, well-motivated, and clearly addresses the research questions. It is supported by comprehensive experimental results, ablation studies, scaling analysis, and domain adaptation analysis. Extensive experiments on both short-term and long-term forecasting under few-shot settings demonstrate the effectiveness of FactoST compared to other foundational and expert models. It achieves lower Mean Absolute Error and Root Mean Squared Error with fewer parameters and shorter inference times than other state-of-the-art methods.

Weaknesses

1. What is the contribution of L_pred and L_recon during pretraining? Are there any weights associated with these losses in equation 3 and equation 4?
2. Zero-shot evaluation is a crucial aspect of assessing foundational models, but FactoST has not been evaluated in this setting.
3. The batch size of 16,384 was selected for pretraining to ensure stable optimization. However, what does "stability" specifically refer to in this context? Given the large and computationally intensive nature of this batch size, was the impact of different batch sizes on pretraining and fine-tuning performance explored or evaluated?
4. Abbreviation MAE used (line 11) before its full form Mean Absolute Error (line 547).
5. On line 262 Mean Square Error (MSE) is used instead of MAE.

---

> ### Author Rebuttal · Authors · 2025-07-30
>
> Thank you for recognizing the good motivation, clear design, comprehensive evaluation, and unique innovation of our paper. Below are our responses to your major questions.
>
> > **Q1 (Weakness): What is the contribution of L_pred and L_recon during pretraining? Are there any weights associated with these losses in Equation 3 and Equation 4?**
> >
>
> **A1:** `L_pred` supervises future sequence prediction, capturing both short-term dynamics and long-term trends. `L_recon` enhances multi-frequency pattern modeling through masked frequency reconstruction. The two losses are **complementary**, so we adopt **equal-weighted** joint optimization (`L_pretrain = L_pred + L_recon`) without separate weighting.
>
> To validate their effectiveness, we conducted an ablation study on zero-shot and few-shot long-term forecasting for ETTh2. Results show that removing `L_recon` **degrades** performance: zero-shot MAE/RMSE increase by 3.54% and 5.27%, respectively, while few-shot MAE/RMSE increase by 0.78% and 0.87%. This **highlights** the importance of **self-supervised multi-frequency reconstruction** for learning robust temporal representations.
>
> | **Model Variants** | **Zero-shot (MAE / RMSE)** | **Few-shot (MAE / RMSE)** |
> | --- | --- | --- |
> | FactoST | 0.395 / 0.588 | 0.383 / 0.578 |
> | FactoST w/o L_recon | 0.409 / 0.619 | 0.386 / 0.583 |
> | Δ (%) MAE / RMSE | +3.54% / +5.27% | +0.78% / +0.87% |
>
> > **Q2 (Weakness): Zero-shot evaluation is a crucial aspect of assessing foundational models, but FactoST has not been evaluated in this setting.**
> >
>
> **A2:** Zero-shot evaluation is indeed crucial and **has been performed**. Comprehensive results are presented in **Appendix A.4 (Table 6)**, covering both short- and long-term forecasting. We further discuss the **limitation** of the zero-shot capability in Q4/A4 of Reviewer rPHb and propose some **future directions** in Q2/A2 of Reviewer jNeH.
>
> > **Q3 (Weakness): The batch size of 16,384 was selected for pretraining to ensure stable optimization. However, what does "stability" specifically refer to in this context? Given the large and computationally intensive nature of this batch size, was the impact of different batch sizes on performance explored?**
> >
>
> **A3:** "Stability" refers to a balance between **training speed** and the **observability** of the optimization process. We tested batch sizes of 8,192, 16,384, and 24,576. All converged to similar loss values with **no significant impact** on final downstream performance. We chose 16,384 because: 8,192 resulted in too many iterations (slow convergence), while 24,576 resulted in too few iterations, making it difficult to monitor loss changes. Thus, 16,384 provided the optimal balance for managing the training process.
>
> > **Q4 (Weakness - Typo/Misc): Abbreviation MAE is used (line 11) before its full form "Mean Absolute Error" (line 547). On line 262, "Mean Square Error (MSE)" is used instead of MAE.**
> >
>
> **A4:** **Corrections have been made in the revised manuscript:** "Mean Absolute Error (MAE)" is now defined at its first appearance (line 11), and the typo on line 262 (MSE corrected to MAE) has been fixed. All technical terms are now consistently defined upon first use.
>
> ---
>
> Thank you for your meticulous suggestions. We will address all your comments, including the experimental details and typographical corrections, in the revised manuscript.

---

> > ### Comment · Reviewer_cLs2 · 2025-08-05
> >
> > I thank the authors for addressing my questions in the rebuttal. Most of my concerns regarding the contribution of ```L_pred``` and ```L_recon``` during pretraining, clarification on zero-shot evaluation and stability claims have been satisfactorily resolved. Kindly integrate the justification provided in the rebuttal in the final version of the paper. Therefore, I will raise my current score and support its acceptance.

---

### Official Review · Reviewer_rPHb · 2025-07-02

**Clarity:** 4
**Significance:** 4
**Originality:** 3
**Rating:** 5
**Confidence:** 5

**Summary:**

This paper proposes a spatio-temporal foundation model, with its training process primarily divided into two stages: universal temporal pretraining and spatio-temporal adaptation, addressing challenges faced by existing spatio-temporal foundation models such as high computational costs.

**Questions:**

1. How does the proposed method effectively handle inputs with different temporal scales, such as 12 and 96 time steps?

2. Does the spatio-temporal adaptation module participate in the pre-training process, or is it only involved in few-shot learning?

3. During pre-training, are all models (including foundation models and expert models) (re)trained using the same pre-training dataset? Or do the foundation models require only few-shot learning?

**Ethical Concerns:**

["NO or VERY MINOR ethics concerns only"]

**Final Justification:**

The concerns raised have been sufficiently addressed, including clarification on how the model handles inputs of varying lengths, as well as detailed explanations of the zero-shot and few-shot experimental settings and results. I consider this to be a high-quality paper, and therefore I will maintain my original score.

**Limitations:**

yes

**Paper Formatting Concerns:**

N.A.

**Quality:**

4

**Strengths And Weaknesses:**

**Strengths**

1. Compared to existing spatiotemporal models, the proposed method maintains excellent prediction performance while ensuring high computational efficiency.

2. The proposed foundation model combines Universal Temporal Pretraining and novel Spatio-Temporal Adaptation components, enabling effective adaptation to few-shot tasks and making it applicable to a wide range of real-world scenarios.

3. The experiments are comprehensive, validating the model's effectiveness through various aspects including few-shot learning, zero-shot learning, scaling laws, efficiency experiment and ablation studies.

**Weaknesses**
1. The analysis of the model's zero-shot experiments is limited, with the model's zero-shot performance being comparable to some existing foundation models. What are the potential reasons for this?

2. In few-shot learning experiments, almost all test dataset task types are included in the pre-training set, which limits the validation of the model's cross-task performance.

3. The experimental configuration details are insufficient. For example, specific configurations for both foundation and non-foundation models during the pre-training phase are not clearly described, and it's unclear how the proposed model adapts to predictions with different time steps.

---

> ### Author Rebuttal · Authors · 2025-07-30
>
> Thank you for recognizing the effectiveness, novel design, and rigorous evaluation of our work. Below we endeavor to address your questions.
>
> > **Q1: How does the proposed method effectively handle inputs with different temporal scales, such as 12 and 96 time steps?**
> >
>
> **A1:** FactoST employs a Transformer encoder-decoder architecture. During the UTP stage, it utilizes both MLP prediction heads and linear reconstruction heads to support multi-horizon forecasting (e.g., 12/96 steps), enabling robust temporal modeling. This flexibility is achieved through patch-based tokenization, where input sequences are divided into non-overlapping patches, allowing adaptive processing of both short-term and long-term inputs. In the subsequent STA stage, only the encoder and prediction heads are fine-tuned. **Horizon-specific prediction heads** are initialized based on the target sequence length, allowing the model to effectively manage varying temporal scales.
>
> > **Q2: Does the spatio-temporal adaptation module participate in the pretraining process, or is it only involved in few-shot learning?**
> >
>
> **A2:** The STA module participates **only** in few-shot fine-tuning and is **explicitly excluded from pretraining**. During UTP, FactoST learns universal temporal patterns without any spatial modeling. As detailed in Section 3.2, spatial information is injected later via lightweight adapters in the STA stage, preserving the pretrained temporal capabilities. This factorized learning approach is further explained in Q1/A1 of Reviewer UUgZ.
>
> > **Q3: During pretraining, are all models (including foundation models and expert models) (re)trained using the same pretraining dataset? Or do the foundation models require only few-shot learning?**
> >
>
> **A3:** No, the models are not retrained on the same pretraining dataset.
>
> - **Foundation Models (e.g., FactoST, UniST, Moirai):** These are pretrained on their respective, distinct cross-domain corpora (e.g., FactoST's UTP uses Monash; UniST uses grid data; TimesFM uses large-scale time series data). They are then evaluated via few-shot adaptation on downstream datasets.
> - **Expert Models (e.g., STID, PatchTST):** These are typically trained from scratch on individual target datasets, without any cross-domain pretraining.
>
> Therefore, foundation models undergo extensive pretraining on their specific corpora and are subsequently adapted (not retrained from scratch) via few-shot learning on target tasks.
>
> > **Q4 (Weakness): The analysis of the model's zero-shot experiments is limited, with the model's zero-shot performance being comparable to some existing foundation models. What are the potential reasons for this?**
> >
>
> **A4:** The zero-shot performance of STFMs is fundamentally limited by **domain-specific spatial heterogeneity**. Notably, FactoST (i.e., UTP without STA) and TimesFM, which **lack explicit spatial learning**, already **outperform** STFMs like OpenCity and UniST in zero-shot scenarios. This validates our core motivation: joint spatio-temporal modeling during pretraining can **hinder generalization** across domains with differing spatial structures.
>
> Furthermore, the pretraining corpora vary significantly in **domain** and **scale**:
>
> | **Model** | **Pretraining Corpus** | **Statistics** |
> | --- | --- | --- |
> | FactoST | Monash (5 domains, 15 datasets) | 13M time points |
> | OpenCity | 21 heterogeneous traffic datasets (10,110 regions) | 151.1M observations |
> | UniST | 21 multi-source grid datasets | - |
> | TimesFM | Large-scale real-world and synthetic time series data | 100B time points |
> | Moirai | LOTSA dataset | 27B time points |
>
> Despite its **significantly smaller scale**, FactoST's zero-shot performance remains **competitive**, demonstrating the effectiveness of our UTP's multi-frequency augmentation, multi-domain prompting, and multi-task pretraining.
>
> We also acknowledge that there is significant room for improvement in zero-shot performance. To this end, we have explored future directions, such as **Test-Time Adaptation (TTA)**, which can further boost zero-shot results. For a detailed discussion on these potential extensions, please see **Q2/A2 of Reviewer jNeH**.
>
> > **Q5 (Weakness): In few-shot learning experiments, almost all test dataset task types are included in the pretraining set, which limits the validation of the model's cross-task performance.**
> >
>
> **A5:** Although the pretraining and few-shot datasets share overlapping semantic domains, they differ substantially in **spatial granularity** (e.g., regions vs. sensors), **temporal resolution** (e.g., hours vs. minutes), and **task objectives** (e.g., pedestrian count vs. traffic speed). This significant difference in data characteristics demonstrates the model's true cross-task generalization ability, even if the high-level semantics overlap. Detailed dataset statistics are provided in Appendix A.1.1.
>
> Furthermore, UTP's multi-frequency augmentation **explicitly enhances** temporal pattern diversity, improving cross-task transferability. The factorized design (UTP for universal patterns, STA for domain-specific adaptation) **inherently** improves cross-task adaptability compared to joint pretraining approaches.
>
> > **Q6 (Weakness): The experimental configuration details are insufficient. For example, specific configurations for both foundation and non-foundation models during the pretraining phase are not clearly described.**
> >
>
> **A6:** Comprehensive experimental configurations for both the UTP and STA stages are detailed in **Appendix A.1.2**, including model architectures, training procedures, and parameter settings (e.g., 3 encoder/decoder layers, 16 attention heads, latent dimension 128). For baseline models, hyperparameters followed the original papers' defaults or were tuned via grid search on validation sets (Appendix A.3). All baselines were evaluated within a **unified** framework using **identical** data splits to ensure **fair comparison**.
>
> ---
>
> Thank you again for your constructive feedback. We will incorporate all the suggested clarifications and improvements into the revised manuscript.

---

> > ### Comment · Reviewer_rPHb · 2025-08-06
> >
> > Thank you for your response, which has addressed most of my concerns very well. However, I still have a few questions I would like to discuss further. It seems that the scaling analysis was conducted primarily from the perspective of data volume. In Figure 4 and the corresponding table, could you clarify what "zero" refers to? Does it indicate the performance of the model with randomly initialized parameters without any training? Additionally, have you conducted any experiments related to model size, such as increasing the number of parameters? I am curious whether scaling up the model could lead to performance improvements. (If not, this would be worth investigating in future work)

---

> > > ### Author Response · Authors · 2025-08-07
> > > **Rebuttal by Authors of Reviewer rPHb**
> > >
> > > Thanks again for your thoughtful feedback. We are pleased to hear that our previous response addressed most of your concerns. Below, we provide further clarification on your remaining questions.
> > >
> > > > **Q7: In Figure 4 and the corresponding table, could you clarify what "zero" refers to?**
> > > >
> > >
> > > "Zero" refers to zero-shot evaluation, where the model is applied directly to downstream datasets without any fine-tuning, relying solely on pretrained knowledge. This figure is designed to evaluate the **rapid adaptation capability** of FactoST’s STA in downstream spatio-temporal tasks under rapidly increasing data volumes. We also include data scaling experiments on the pretraining side of FactoST’s UTP, as discussed in **Q3/A3 of Reviewer UUgZ**.
> > >
> > > > **Q8: Have you conducted any experiments related to model size, such as increasing the number of parameters? I am curious whether scaling up the model could lead to performance improvements.**
> > > >
> > >
> > > Thanks for the suggestions. Due to time constraints, we conducted an ablation study on the number of transformer layers (i.e., `n_layers`) in FactoST’s backbone, with model parameters increasing from 3.0M to 4.3M as depth grows. We evaluated the model on ETTh2 for long-term forecasting, varying `n_layers` from 1 to 7. Results show that zero-shot performance improves from 1 to 3 layers, then plateaus despite further increases in capacity. In the few-shot setting, performance peaks at 3 layers and gradually declines from 5 to 7, with training logs indicating overfitting—likely due to increased depth without adjusting hidden dimensions or dropout rates.
> > >
> > > | n_layers | **Zero-shot (MAE/RMSE)** | **Few-shot (MAE/RMSE)** |
> > > | --- | --- | --- |
> > > | 1 (3.0M) | 0.402 / 0.597 | 0.397 / 0.590 |
> > > | 3 (3.4M) | 0.395 / 0.588 | 0.383 / 0.578 |
> > > | 5 (3.8M) | 0.394 / 0.588 | 0.388 / 0.585 |
> > > | 7 (4.3M) | 0.394 / 0.589 | 0.391 / 0.586 |
> > >
> > > Overall, our results indicate that proper scaling can lead to performance improvements, we will include a more detailed analysis in the final revised version of the paper.

---

> > > > ### Comment · Reviewer_rPHb · 2025-08-08
> > > >
> > > > Thank you for your response; it has cleared up all my concerns.

---

### Note · Authors · 2025-08-12

We sincerely thank all reviewers, Area Chairs, and Program Chairs for their insightful feedback, which has greatly enhanced the rigor and clarity of our work. We are glad that all reviewers have confirmed their concerns are resolved and expressed support for the paper's acceptance, and we will fully integrate these discussions and clarifications into the revised manuscript.

We believe FactoST offers a practical path toward scalable, generalizable spatio-temporal foundation models, with implications for real-world applications in traffic, energy, environmental forecasting, and more.

Thank you again for your time and guidance.

Sincerely,
All 5129 Authors

---

### Decision · Program_Chairs · 2025-09-17

**Decision:**

Accept (spotlight)

**Comment:**

This paper introduces FactoST, a novel two-stage foundation model for spatio-temporal forecasting. The core contribution is a "factorized" paradigm that decouples universal temporal pre-training from spatio-temporal adaptation. This approach amis to circumvent the high computational costs and spatial generalization challenges inherent in existing models.

All reviewers support the acceptance of the paper, recognizing it as a strong contribution.

Strengths:
* novelty and impact: the central idea of decoupling temporal and spatial learning is conceptually clear, novel, and potentially influential for future research in spatio-temporal foundation models (jNeH, UUgz, cLs2).
* strong empirical performance: the results significantly outperform prior models and remain competitive with expert models in few-shot scenarios (rPHb, cLs2)
* comprehensive evaluation: the paper is supported by extensive experiments, including detailed ablation, scaling analysis, domain adaptation visualization (rPHb, cLs2)

Weaknesses:
Initial reviews raised several points for clarification. these include dthe scorep of the term "factorized" (UUgZ), the need for more baselines and zero-shot evaluation (cLs2, UUgZ), the generalizability of the adaptation module, details on handling different data scales and potential imbalances (rPHb, jNeH).

The authors provided a comprehensive and convincing rebuttal that successfully addressed all major concerns.
Hence, I recommend an acceptance for this paper.